

# Recent updates on the Copernicus Marine Service global ocean monitoring and forecasting real-time 1/12° high resolution system

Jean-Michel Lellouche[1], Eric Greiner[2], Olivier Le Galloudec[1], Gilles Garric[1], Charly Regnier[1], Marie Drevillon[1], Mounir Benkiran[1], Charles-Emmanuel Testut[1], Romain Bourdalle-Badie[1], Florent Gasparin[1], Olga Hernandez[1], Bruno Levier[1], Yann Drillet[1], Elisabeth Remy[1], Pierre-Yves Le Traon[1,3]

[1] Mercator Ocean, Ramonville Saint Agne, France

[2] Collecte Localisation Satellites, Ramonville Saint Agne, France

[3] IFREMER, 29280, Plouzané, France

*Correspondence to*: Jean-Michel Lellouche (jlellouche@mercator-ocean.fr)

**Abstract**

Since October 19, 2016, and in the framework of Copernicus Marine Environment Monitoring Service (CMEMS), Mercator Ocean delivers in real-time daily services (weekly analyses and daily 10-day forecasts) with a new global 1/12° high resolution (eddy-resolving) monitoring and forecasting system. The model component is the NEMO platform driven at the surface by the IFS ECMWF atmospheric analyses and forecasts. Observations are assimilated by means of a reduced-order Kalman filter with a 3D multivariate modal decomposition of the forecast error. Along track altimeter data, satellite sea surface temperature, sea ice concentration and in situ temperature and salinity vertical profiles are jointly assimilated to estimate the initial conditions for numerical ocean forecasting. A 3D-VAR scheme provides a correction for the slowly-evolving large-scale biases in temperature and salinity.

This paper describes the recent updates applied to the system and discusses the importance of fine tuning of an ocean monitoring and forecasting system. It details more particularly the impact of the initialization, the correction of precipitation, the assimilation of climatological temperature and salinity in the deep ocean, the construction of the forecast error covariance





and the adaptive tuning of observations error on increasing the realism of the analysis and
forecasts.
The scientific assessment of the ocean estimations are illustrated with diagnostics over some
particular years, assorted with time series over the time period 2007-2016. The overall impact
of the integration of all updates on the products quality is also discussed, highlighting a gain
in performance and reliability of the current global monitoring and forecasting system
compared to its previous version.
## 1  Introduction
Mercator Ocean monitoring and forecasting systems have been routinely operated in real-time
since early 2001 and have been regularly upgraded by increasing complexity, expanding the
geographical coverage from regional to global and improving models and assimilation
schemes (Brasseur et al., 2006; Lellouche et al., 2013).
After having successfully coordinated the European MyOcean and MyOcean2 projects
(http://www.myocean.eu), Mercator Ocean was officially entrusted by the European
Commission on November, 11, 2014 to implement and operate the Copernicus Marine
Environment Monitoring Service (CMEMS), as part of the European Earth observation
program Copernicus (http://marine.copernicus.eu). Since January 2009, Mercator Ocean,
which had primary responsibility for the global ocean forecasts of the MyOcean project,
developed several versions of its monitoring and forecasting systems for the various
milestones (from V0 to V4) of the MyOcean project,  and more recently, for milestones V1,
V2 and V3 of the CMEMS (Fig. 1). The main differences and links between the various
versions of the Mercator Ocean systems in the framework of past MyOcean project and
current CMEMS are summarized in Table 1 an Table 2 for Intermediate Resolution ¼° Global
configurations (hereafter IRG) and High Resolution 1/12° Global configurations (hereafter
HRG) systems respectively.
These systems are intensively used in four main areas of application: maritime safety, marine
resources management, coastal and marine environment, and weather, climate and seasonal
forecasting (http://marine.copernicus.eu/markets/use-cases). As described in Lellouche et al.
(2013), the evaluation of such systems includes routine verification against assimilated and
independent in situ and satellite observations, as well as a careful check of many physical
processes (e.g. mixed layer depth evaluation as shown in Drillet et al. (2014)). Scientific





studies brought precious additional evaluation feedbacks (Juza et al., 2015; Smith et al., 2016;
Estournel et al., 2016). Finally, several studies showed the added value of surface currents
analyses provided by these systems for drift applications (Scott et al., 2012; Drevillon et al.,

4    2013).

Since May 2015, Mercator Ocean opened the CMEMS and has been in charge of the global
high resolution ocean analyses and forecasts. In this context, R&D activities have been
conducted these last years to improve the real-time 1/12° high resolution (eddy-resolving)
global analysis and forecasting system. Since October 19, 2016, Mercator Ocean delivers in
real-time daily services (weekly analyses and daily 10-day forecasts) with a new global 1/12°
system PSY4V3R1 (hereafter PSY4V3, and corresponding to HRG_V2V3 in Fig. 1). Note
that PSY4V3 will be the system for the CMEMS V4 milestone. In this system, the ocean/sea
ice model and the assimilation scheme benefit of the following main updates: atmospheric
forcing fields are corrected at large-scale with satellite data; freshwater runoff from ice sheets
melting is added to river runoffs; a time varying global average steric effect is added to the
model sea level; the  last version of GOCE geoid observations are taken into account in the
Mean Dynamic Topography used for Sea Level Anomalies assimilation; adaptive tuning is
used on some of the observational errors; a dynamic height criteria is added to the Quality
Control of the assimilated temperature and salinity vertical profiles; satellite sea ice
concentrations are assimilated; climatological temperature and salinity in the deep ocean are
assimilated below 2000 m to prevent drifts in those very sparsely observed depths.
The impact of all these updates can be evaluated separately, thanks to an incremental
implementation, taking advantage of Mercator Ocean's specific hierarchy of system
configurations running with identical set up. To this aim, short simulations (from one year to a
few years) were performed by adding from one simulation to another one upgrade at a time,
using the IRG configuration or some high resolution 1/12° regional configuration.
Moreover, in the development phase of an operational system, it was decided to
systematically perform three twin numerical simulations over a given time period,
maintaining the same ocean model tunings but varying the complexity and the level of data
assimilation. Inter-comparisons between the three simulations were then conducted in order to
better analyze and to try to quantify the impact of some component of the assimilation system.
These three versions of system have also been used to quantify the impact of some updates.
In a previous paper (Lellouche et al., 2013), the main results of the scientific evaluation of
MyOcean global monitoring and forecasting systems at Mercator Ocean showed how





refinements or adjustments to the system impacted the quality of ocean analyses and
forecasts. The primary objective of this paper is to describe the recent updates applied to the
system PSY4V3. The updates showing the highest impact on the products quality are
separately illustrated and discussed, with a particular focus on the initialization, the correction
of precipitation, the assimilation of climatological temperature and salinity in the deep ocean,
the construction of the forecast error covariance and the adaptive tuning of observations error.
Another objective of this paper is to present a first level evaluation of the system. The purpose
here is not to perform an exhaustive validation but only to check the global behavior of the
system compared to assimilated quantities or independent observations. Thus, an assessment
of the hindcasts (2007-2016) quality is conducted and improvements with respect to the
previous system are highlighted in order to show the level of performance and the reliability
of the system PSY4V3. A complementary study aimed at demonstrating the scientific value of
PSY4V3 for resolving oceanic variability at regional and global scale (Gasparin et al., 2018 -
Submitted in Journal of Marine Systems). Lastly, several scientific studies have investigated
local ocean processes by comparing the PSY4V3 system with independent observations
campaigns (Koenig et al., 2017; Artana et al., 2018 - Submitted in Journal of Geophysical
Research - Oceans). This reinforces the system PSY4V3 evaluation effort.
This paper is organized as follows. The main characteristics of the system PSY4V3 and
details concerning the updates are described in Sect. 2. The impact of the most sensitive
upgrades is shown in Sect. 3. Results of the scientific evaluation, including some comparisons
with independent observations, are given in Sect. 4. Section 5 contains a summary of the
scientific assessment, as well as a discussion of the future improvements for the next version
of the global high resolution system.

## 2   Description of the current global high resolution monitoring and forecasting system PSY4V3

This section contains the main characteristics of the CMEMS system PSY4V3 and details the
last updates to the system compared to the previous system PSY4V2R2 (hereafter PSY4V2,
and corresponding to HRG_V3V4_V1V2 in Fig. 1). A detailed description of the main
updates is provided in Sect. 3.



## 2.1  Physical model and latest updates

The system PSY4V3 uses version 3.1 of the NEMO ocean model (Madec et al., 2008). This NEMO version is available since a few years and has been already used in the previous system PSY4V2. However, all the schemes and the parameterizations used in this version are still available in the current NEMO 3.6 stable version that is now the standard version of the code. The physical configuration is based on the tripolar ORCA12 grid type (Madec and Imbard, 1996) with a horizontal resolution of 9 km at the equator, 7 km at Cape Hatteras (mid-latitudes) and 2 km toward the Ross and Weddell seas. The 50-level vertical discretization retained for this system has a decreasing resolution from 1m at the surface to 450 m at the bottom, and 22 levels within the upper 100 m. A "partial cells" parameterization (Adcroft et al., 1997) is chosen for a better representation of the topographic floor (Barnier et al., 2006) and the momentum advection term is computed with the energy and enstrophy conserving scheme proposed by Arakawa and Lamb (1981). The advection of the tracers (temperature and salinity) is computed with a total variance diminishing (TVD) advection scheme (Levy et al., 2001; Cravatte et al., 2007). We use a free surface formulation. External gravity waves are filtered out using the Roullet and Madec (2000) approach. A laplacian lateral isopycnal diffusion on tracers (100 $m^2$ $s^{-1}$) and a horizontal biharmonic viscosity for momentum (-2e10 $m^4$ $s^{-1}$) are used. In addition, the vertical mixing is parameterized according to a turbulent closure model (order 1.5) adapted by Blanke and Delecluse (1993), the lateral friction condition is a partial-slip condition with a regionalization of a no-slip condition (over the Mediterranean Sea) and the Elastic-Viscous-Plastic rheology formulation for the LIM2 ice model (Fichefet and Maqueda, 1997) has been activated (Hunke and Dukowicz, 1997). Instead of being constant, the depth of light extinction is separated in Red-Green-Blue bands depending on the chlorophyll data distribution from mean monthly SeaWIFS climatology (Lengaigne et al., 2007). The bathymetry used in the system is a combination of interpolated ETOPO1 (Amante and Eakins, 2009) and GEBCO8 (Becker et al., 2009) databases. ETOPO1 datasets are used in regions deeper than 300 m and GEBCO8 is used in regions shallower than 200 m with a linear interpolation in the 200 - 300 m layer. Internal-tide driven mixing is parameterized following Koch-Larrouy et al. (2008) for tidal mixing in the Indonesian Seas, as the system doesn't represent explicitly the tides. The atmospheric fields forcing the ocean model are taken from the ECMWF (European Centre for Medium-Range Weather Forecasts) IFS (Integrated Forecast System). A 3 h sampling is used to reproduce the diurnal cycle. Momentum and heat turbulent surface fluxes are computed from the Large and Yeager (2009)




bulk formulae using the following set of atmospheric variables: surface air temperature and
surface humidity at a height of 2 m, mean sea level pressure and wind at a height of 10 m.
Downward longwave and shortwave radiative fluxes and rainfall (solid + liquid) fluxes are
also used in the surface heat and freshwater budgets. Compared to the previous HRG system
PSY4V2, the following updates were done on the model part (see Table 2):
- The bathymetry used in the system benefited from a specific correction in the Indonesian
Sea inherited from the INDESO system (Tranchant et al., 2016).
- In order to solve numerical problems induced by the use of z-coordinates on the vertical
(Willebrand et al., 2001), a relaxation toward the World Ocean Atlas 2013 (version 2)
2005-2012 time period (hereafter WOA13v2,
https://data.nodc.noaa.gov/woa/WOA13/DOC/woa13v2_changes.pdf) temperature
(Locarnini et al., 2013) and salinity (Zweng et al., 2013) climatology has been added at
Gibraltar and Bab-el-Mandeb straits. For Gibraltar (respectively Bab-el-Mandeb), the
relaxation area is centered at 8° W, 35° N (respectively 46° E, 12° N). At the center the
relaxation time is 10 days (respectively 50 days). This time is increased up to infinity 4°
(respectively 5°) away from the center. The relaxation is not constant over the vertical. It
is only applied below 500 m and it is increased linearly between 500 to 700 m. Between
700 m and the bottom of the ocean the coefficient value is unchanged.
- Surface wind stress computation should in principle consider wind speed relative to the
surface ocean currents (Bidlot, 2012; Renault et al., 2016). However, this statement
applies to a fully coupled ocean/atmosphere system, which is not the case for the present
system PSY4V3. Based on sensitivity experiments, we pragmatically consider only 50 %
of the surface model currents in the wind stress computation.
- The monthly runoff climatology is built with data on coastal runoffs and 100 major rivers
from the Dai et al. (2009) database (instead of Dai and Trenberth (2002) for the system
PSY4V2). This database uses new data, mostly from recent years, streamflow simulated
by the Community Land Model version 3 (CLM3) to fill the gaps, in all lands areas except
Antarctica and Greenland. In addition, we built mean seasonal freshwater fluxes
representing Greenland and Antarctica ice sheets and glaciers runoff melting. For this
purpose we have distributed IPCC-AR13 (Church et al., 2013) mean values, 1.51 mm yr$^{-1}$
for Greenland and 6.65 mm yr$^{-1}$ for Antarctica, onto a domain varying seasonally and
defined by the Altiberg icebergs database project (Tournadre et al., 2013). Domain
covered by giant icebergs from Silva et al. (2006) completes southern most areas not
covered by Altiberg data. One third of these quantities is applied off shore and two third





along Greenland and Antarctic coastlines. We also used negative gridded GRACE
anomalies (Bruinsma et al., 2010) to distribute spatially these runoffs along coastlines.
-  As the Boussinesq approximation is applied to the model equations, conserving the ocean
volume and varying its mass, the simulations do not properly directly represent the global
mean steric effect on the sea level (Greatbatch, 1994). For improved consistency with
assimilated satellite observations of sea level anomalies, which are unfiltered from the
global mean steric component, a time-evolving global average steric effect is added to the
sea level in the simulation. This global average steric effect has been computed as the
difference between two successive daily global mean dynamic heights (vertical
integration, from the surface to the bottom, of the specific volume anomaly).
-  Due to large known biases in precipitations, a satellite-based large-scale correction of
precipitations has been performed, except at high latitudes (poleward of 65° N and 60° S).
This is detailed in Sect. 3.
-  In order to avoid mean sea-surface-height drift due to the large uncertainties in the water
budget closure, the following treatments were applied:

o  The surface freshwater global budget is set to an imposed seasonal cycle (Chen et

al., 2005). Only spatial departures from the mean global budget are kept from the

forcing.

o  A trend of 2.2 mm yr$^{-1}$ has been added to the surface mass budget in order to

represent the recent estimate of the global mass addition to the ocean (from

glaciers, land water storage changes, Greenland and Antarctica ice sheets mass

loss) (Chambers et al., 2017). This term is implemented as a surface freshwater

flux in the open ocean domain infested by observed icebergs.

**2.2   Data assimilation and latest updates**
The data are assimilated by means of a reduced-order Kalman filter derived from a SEEK
filter (Brasseur and Verron, 2006), with a 3D multivariate modal decomposition of the
forecast error and a 7-day assimilation cycle. It includes an adaptive-error estimate and a
localization algorithm. This data assimilation system is called SAM (Système d'Assimilation
Mercator). The forecast error covariance is based on the statistics of a collection of 3D ocean
state anomalies, typically a few hundreds (250 anomalies for PSY4V3). The anomalies are
computed from a long numerical experiment (9 years for PSY4V3) with respect to a running
mean in order to estimate the 7-day scale error on the ocean state at a given period of the year.





A Hanning low-pass filter is used to create the running mean with a cut-off frequency equal to
$1/24$ days$^{-1}$. Altimeter data, in situ temperature and salinity vertical profiles, and satellite sea
surface temperature and sea ice concentration are jointly assimilated to estimate the initial
conditions for numerical ocean forecasting. In addition, a 3D-VAR scheme provides a
correction for the slowly-evolving large-scale biases in temperature and salinity (Lellouche et
al., 2013).
Compared to the previous HRG system PSY4V2, the following updates were done on the data
assimilation part (see Table 2):
-    CMEMS         satellite         sea         ice         concentration         OSI         SAF

(http://marine.copernicus.eu/documents/QUID/CMEMS-OSI-QUID-011-001to007-

009to012.pdf) is a new observation assimilated in the system PSY4V3. For this, a separate

monovariate/monodata analysis is carried out for the ice variables, in parallel to that for

the ocean. The two analyses are completely independent.

-    CMEMS            OSTIA            SST            (delayed            time:

http://marine.copernicus.eu/documents/QUID/CMEMS-OSI-QUID-010-011.pdf,       then

near   real   time:   http://marine.copernicus.eu/documents/QUID/CMEMS-OSI-QUID-010-

001.pdf) is assimilated in the system PSY4V3, instead of near real time AVHRR SST

from NOAA in PSY4V2. A particular attention has been devoted to the computation of

the model equivalent. As OSTIA provides the foundation SST (considered nominally at

10 m depth), the SST model equivalent is performed by calculating the night-time average

of the first level of the model temperature.

-    In addition to the quality control based on temperature and salinity innovation statistics

(detection of spikes, large biases), already present in the previous system, a second quality

control has been developed and is based on dynamic height innovation statistics (detection

of small vertically constant biases). This is detailed in Sect. 2.3.

-    A new hybrid MDT, based on the "CNES-CLS13" MDT (Rio et al., 2014) with

adjustments made using the Mercator GLORYS2V3 (GLobal Ocean ReanalYsis and

Simulation – stream 2 – version 3) reanalysis and with an improved Post Glacial Rebound

(also called Glacial Isostatic Adjustment), has been used. This new hybrid MDT also takes

into account the last version of the GOCE geoid. This replaces the previous hybrid MDT

used in the previous system PSY4V2, which was based on the "CNES-CLS09" MDT

derived from observations (Rio et al., 2011). The new hybrid MDT significantly reduces

(not shown) sea level bias (more than 5 cm in some areas) and consequently temperature



and salinity in regions where the topography makes difficult the mean sea surface
estimation (e.g. Indonesia, Red Sea and Mediterranean Sea).
-   A consistent along track SLA dataset
(http://marine.copernicus.eu/documents/QUID/CMEMS-SL-QUID-008-032-051.pdf),
with a 20-year altimeter reference period, is assimilated all along the simulation
performed with the system PSY4V3.
-   The CORA 4.1 CMEMS in situ database (Szekely et al., 2016;
http://marine.copernicus.eu/documents/QUID/CMEMS-INS-QUID-013-001b.pdf) has
been assimilated for the 2006-2013 period. In addition to Argo and other in situ data sets,
this database includes temperature and salinity vertical profiles from sea mammal
(elephant seals) database (Roquet et al., 2011) to compensate for the lack of such data at
high latitudes. From 2014 to present, the near-real time CMEMS product
(http://marine.copernicus.eu/documents/QUID/CMEMS-INS-QUID-013-030-036.pdf) is
assimilated.
-   As the prescription of observation errors in the assimilation systems is not sufficiently
accurate, adaptive tuning of observation errors for the SLA and SST has been
implemented. The method has been adapted from diagnostics proposed by Desroziers et
al. (2005) and is detailed in Sect. 3.
-   New 3D observation errors files for the assimilation of in situ temperature and salinity
data have been re-computed from the MyOcean IGR system PSY3V3R3 (corresponding
to IRG_V3V4 in Fig. 1) using an offline version of the adaptive tuning method mentioned
above.
-   A weak constraint towards the WOA13v2 climatology on temperature and salinity in the
deep ocean (below 2000 m) has been included in the two components (3D-VAR and
SEEK filter) of the assimilation scheme to prevent drifts in temperature and salinity and as
a consequence to obtain a better representation of the sea level trend at global scale in the
system. The method consists in assimilating vertical climatological profiles of temperature
and salinity at large scale and below 2000 m in regions drifting away from the
climatological values, using a non-Gaussian error at depth. This is detailed in Sect. 3.
-   The time window for the 3D-VAR bias correction was reduced from 3 to 1 month to
obtain a correction that is more in line with the current physics, which is made possible by
the good spatial and temporal distribution of the Argo network from 2006.
-   In the previous system PSY4V2, the SSH increment was the sum of barotropic and
baroclinic (dynamic) height increments. Dynamic height increment was calculated from




the temperature and salinity increments, while the barotropic increment was an output of
the analysis. In the system PSY4V3, we directly use the total SSH increment given by the
analysis to take into account, among other things, the wind effect like the hydraulic
control near the straits (Song, 2006; Menemenlis et al., 2007).
-  The uncertainties in the MDT estimate and the sparsity of the observation networks (both
altimetry and in situ profiles) on the 7-day assimilation window do not allow to accurately
estimate the observed global mean sea level. Moreover, the mean sea level time evolution
is the result of an imposed trend for mass inputs (2.2 mm yr$^{-1}$, see Sect. 2.1) together with
a diagnostic steric effect re-computed from model T and S. Therefore, the global mean
increment of the total sea surface height is set to zero and the mean sea level is not
controlled by data assimilation.
-  The error covariance matrices needed for data assimilation are defined using anomalies of
the different variables coming from a simulation in which only a 3D-VAR large scale bias
correction of T, S has been performed (instead of using a free run  as was done in the
previous system PSY4V2). Moreover, these anomalies are spatially filtered in order to
retain only the effective model resolution and in order to avoid injecting noise in the
increments. This is detailed in Sect. 3.
**2.3   Additional Quality Controls on in situ observations**
To minimize the risk of erroneous observations being assimilated in the model, the system
PSY4V3 carries out two successive Quality Controls (QC1 and QC2) on the assimilated T
and S vertical profiles. These are done in addition to the quality control procedures performed
by the data producers.
**2.3.1   Quality Control QC1**
The first quality control QC1 has been already described in Lellouche et al. (2013) and can be
summarized as follows. An observation is considered suspicious if the two following
conditions are both satisfied:

$$\begin{cases} |innovation| > threshold \\ |observation - climatology| > 0.5 * |innovation| \end{cases} \quad (1)$$

where the spatially and seasonally varying *threshold* value comes from statistics (mean,
standard deviation) computed with the very large number of temperature and salinity



innovations collected in the Mercator GLORYS2V1 (GLobal Ocean ReanalYsis and
Simulation – stream 2 – version 1) reanalysis (1993-2009). This first QC allows the detection
of spikes and large biases.
**2.3.2 Quality Control QC2**
The second quality control QC2 is based on dynamic height innovation (vertical integration
from the surface to the bottom) statistics and allows detecting small biases which are present
on the whole water column, and thus can induce large errors. It basically says that the thermal
or haline component of dynamic height innovation ($hdyn(innov_T)$ or $hdyn(innov_S)$) cannot
exceed some threshold in height ($threshold_T$ for thermal component or $threshold_S$ for haline
component). It can be summarized as follows. A vertical profile is rejected if the following
condition is satisfied:

$$\begin{cases} \textbf{For temperature}: \dfrac{|C*hdyn(innov_T)|}{\sum dz_T} > threshold_T \\ \textbf{For salinity}: \dfrac{|C*hdyn(innov_S)|}{\sum dz_S} > threshold_S \end{cases} \tag{2}$$

where
$$\begin{cases} C = 200/\sum dz \quad if \quad 0 < \sum dz \le 200 \\ C = 500/\sum dz \quad if \quad 200 < \sum dz \le 500 \\ C = \sum dz \quad if \quad \sum dz > 500 \end{cases} \tag{3}$$

and $dz_T$ is the model layer thickness corresponding to the temperature observation (same for
$dz_S$ and salinity). These last conditions (Eq. (3)) prevent the threshold from being reached too
quickly in shallow areas.
The average and standard deviation of the thermal or haline components of dynamical height
innovation have been calculated from a global simulation at 1/4°, which is a twin simulation
of the PSY4V3 one. Note that the simulation at 1/4° also assimilates the CORA 4.1 CMEMS
in situ database. The temperature and salinity threshold 2D fields used by QC2 are then
computed as the average plus six times the standard deviation of the dynamical height
innovations (Fig. 2). With these temperature and salinity thresholds, the system will reject
more easily biased salinity profiles in the tropics and biased temperature profiles in strong
currents.
It should also be noted that the QC2 quality control rejects the entire vertical profile while the
QC1 quality control only rejects aberrant temperature and/or salinity values at some given
depths on the vertical profile.



Figure 3a shows an example of a "wrong" temperature profile detected by the QC2 (and not
by the QC1) at the end of July 2008. In this case, $threshold_T$ is equal to 0.3 m (Fig. 3b). The
first condition of Eq. (2) is satisfied and the profile is rejected. When this profile is
assimilated (simulation without QC2), abnormal temperature RMS innovation values appear
at the temporal position (July 2008) of this profile in the Azores region (Fig. 3c). Using QC2
quality control allows solving the problem for this particular profile but also for some others
profiles (see Fig. 3c).
Statistics of the QC1 and QC2 quality controls are summarized in Fig. 4, where the
percentage of suspicious temperature and salinity profiles is given as a function of the year
over the 2007-2016 period. This percentage is relatively stable for both temperature and
salinity profiles, with little year-to-year variability, except for the years 2012 and 2013 where
more suspicious temperature and salinity profiles than usual were detected. Nevertheless, this
percentage remains relatively low (less than 0.35 % for temperature and 3.5 % for salinity),
knowing that the number of temperature profiles available each year ranges between 1.1
million and 1.7 million and the number of salinity profiles between 150,000 and 600,000.

## 17   3    Impact of major updates

Most of the deficiencies in the systems can be related to these main recurring problems:
initialization, atmospheric forcing biases, abyssal circulation and efficiency of the
assimilation schemes. The first three problems are related to uncertainties in poorly observed
areas or parameters (i.e. deep ocean, ice thickness) and to intrinsic errors of the atmospheric
forcing. The last problem is related to linearity and stationarity hypotheses in the assimilation
schemes. In this section, we detail the solutions adopted for the system PSY4V3, reducing
uncertainties in the thermohaline component and allowing flow dependence in our
assimilation scheme.

### 26   3.1    Initialization of oceanic simulation

One way to initialize physical ocean model simulations is by using climatological values of
temperature and salinity from databases and assuming the velocity field is zero at the start.
The model physics then spins up a velocity field in balance with the density field. Another
common way to initialize a model is with fields from a previous run of that model, or with the
results from another model.



Given that data assimilation of the current observation network rapidly (in about 3 months)
adjusts the model state in the first 1000 m, the first solution has been chosen to avoid potential
drifts occurring after some years of simulation. Compared with the previous system PSY4V2
starting in October 2012 from the WOA09 3D climatology (see Fig. 1), the PSY4V3 system
starts in October 2006 using improved initial climatological conditions. For that, we chose to
use ENACT-ENSEMBLES EN4 1° global product (Good et al., 2013) which consists in
monthly objective analyses. The great interest of these monthly fields is that a 3D observation
weight (between 0 and 1) describes the influence of the observations for each field. This
information helps to retain only the observed points and not the perpetual climatology. This
allows the computation of validated trends for each month and of climatology for a particular
date. For that, a pointwise linear regression and in particular the Kendall's robust line-fit
method (Hoaglin et al., 1983) is used, allowing us to obtain an initial condition called "robust
EN4" for any time based only on real observations.
Two free simulations (without any data assimilation) have been performed with the system
PSY4V3, using either WOA09 or robust EN4 as initial condition in October 2006. Figure 5
shows the box-averaged innovations of temperature and salinity as a function of time and
depth over the October 2006 - December 2007 period. The top left panel reveals that, using
WOA09 as initial condition, a fresh bias appears in the first 100 meters of the innovation,
particularly more pronounced at the surface. It is not anymore the case when using robust
EN4 to initialize the model (top right panel). For temperature, the bottom left panel exhibits
cold biases above 100 m and below 300 m that are considerably reduced by using robust EN4
as initial condition (bottom right panel). The warm bias between 200 m and 300 m is slightly
reinforced but it concerns only the top 300 m and this will be corrected by the assimilation of
Argo profiles. Deeper biases are reduced with this new initialization.
**3.2   Correction of precipitations**
Many studies (e.g. Janowiak et al., 1998; Janowiak et al., 2010; Kidd et al., 2013) have
compared reanalysis and atmospheric model precipitation fields with observation-based
datasets, and have shown that atmospheric model products always bring significant and
systematic errors, and are not able to close the global average freshwater budget. For instance,
Janowiak et al. (2010) found that the IFS operational model and ERA-Interim reanalysis (Dee
et al., 2011) from ECMWF perform well for temporal variability with respect to observational
datasets, but they globally overestimate the daily precipitations. Although progresses have





been made in the ECMWF forecast model, substantial errors still occur in the tropics (Kidd et
al., 2013). The correction of atmospheric forcing within ocean applications has already been
successfully explored by adjusting atmospheric fluxes via observational datasets in global
applications (Large and Yeager, 2009; Brodeau et al., 2010). Other studies only focused on
precipitation correction (Troccoli and Kallberg, 2004; Storto et al., 2012).
The proposed method in this paper consists of correcting the daily precipitation fluxes by
means of a monthly climatological coefficient, inferred from the comparison between the
Remote Sensing Systems (RSS) Passive Microwave Water Cycle (PMWC) product (Hilburn,
2009) and the IFS ECMWF precipitations. We use remote PMWC product because of its
relative high 1/4° resolution able to represent more accurately narrow permanent features such
as the Intertropical Convergence Zone. The use of spatially varying monthly climatological
coefficient is justified by the fact that the inter-annual variability is well captured by the
ECMWF forecast model and allows us to apply the correction outside the special sensor
microwave/imager era. This latter assertion is a limitation of the method as it assumes the
operational ECMWF forecast model has a constant bias. In order to avoid discontinuities
when either PMWC or ECMWF products exhibit zero precipitation, e.g. in arid areas, we do
not apply any correction in monthly mean values less than 1 mm of rainfalls fluxes. Also, in
order to keep the more accurate small-scale signal from the high resolution forcing, the
correction is only applied to large-scale component obtained by a low-pass Shapiro filter.
Hilburn et al. (2014) provided accuracy of RSS over ocean rain retrievals validated against
well established long-term in situ datasets such as observations from Pacific Marine
Environment Laboratory rain gauges on moored buoys in the tropics. They found that on
monthly averages, the standard deviation between satellite and buoy is 15.5 %. The
differences are greatest in the Indian Ocean and Western Pacific. We then arbitrarily capped
the correction beyond 20 % in order to take into account these satellite-based retrievals errors.
Lastly, we did not apply the correction poleward 65° N and 60° S because of lack and
important biases of satellite-based precipitations estimate (Lagerloef et al., 2010) at high
latitudes.
Figure 6 represents the difference between the IFS precipitations coming from ECMWF and
the PMWC product using satellite data, before and after large scale correction. Original IFS
forcings exhibit a systematic over-estimation of precipitation within the inter-tropical
convergence zones (up to 3 mm day$^{-1}$) and under-estimation at mid- and high-latitudes (up to
−4 mm day$^{-1}$). After correction, the mean bias compared with PMWC is reduced from 0.47 to
0.19 mm day$^{-1}$.

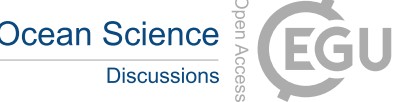



To validate this correction, two global ocean hindcast simulations of several years, using only
the 3D-VAR large-scale biases correction in temperature and salinity, have been performed,
one with IFS correction and the other without. Figure 7 represents the mean surface salinity
innovation (difference between the assimilated observation and the model) on the year 2011.
These maps demonstrate that the IFS correction is beneficial in many areas, reducing the
magnitude of the near-surface salinity fresh mean bias in the Tropics down to 0.5 psu.
## 3.3 Assimilation of climatological temperature and salinity climatology in the
deep ocean
Due to unresolved processes (internal waves, spurious mixing in overflow regions, tidal
mixing) and inaccurate atmospheric forcing (bulk formulas), the model may drift at depth.
Unfortunately, there are very few temperature and salinity profiles below 2000 m to constrain
the model drift. Hence, the climatology is currently the only source of information at depth to
prevent the model from drifting. Virtual vertical profiles of temperature and salinity below
2000 m are built from the monthly WOA13v2 climatology. These virtual observations are
geographically positioned on the model horizontal grid with a coarse resolution (1° x 1°) and
on the model vertical levels from 2200 m to the bottom.
As in Greiner et al. (2006), we define empirically the standard deviations (departures from the
climatology) $\sigma_T$ for temperature and $\sigma_S$ for salinity, as a simple linear vertical profile:

$$
\begin{cases}
\sigma_T = MAX\left(\left(\frac{0.6 - z/10^4}{3}\right); 0.05\right) \\
\sigma_S = \sigma_T/8
\end{cases}
\tag{4}
$$

where $z$ is the depth (in meters).
We define then $\sigma_{TS}$ the density departure from the climatology:

$$
\sigma_{TS} = \alpha\, \sigma_T + \beta\, \sigma_S
\tag{5}
$$

where $\alpha$ represents the thermal expansion coefficient and $\beta$ the saline contraction coefficient.
Following Jackett and Mcdougall (1995), these coefficients are assumed to depend only on
latitude and depth of the ocean as illustrated by Fig. 8.
If we note $d_{TS}$ the density innovation, $d$ the temperature or the salinity innovation and $\sigma$ the
temperature or the salinity departure from the climatology, the value of the climatological
error $e$ is prescribed as:



$$
\begin{cases}
If \quad |d_{TS}| \leq 2\,\sigma_{TS} \quad then \quad e = \infty \ (observation\ rejected) \\
If \quad |d_{TS}| > 2\,\sigma_{TS} \quad then \begin{cases} if \quad 2\sigma < |d| < 3\sigma \quad then \quad e = MIN\left(\frac{2\sigma}{3}\left(\frac{|d|}{|d|-2\sigma}\right); 20\sigma\right) \\ if \quad |d| \geq 3\sigma \ then\ e = 2\sigma \\ if \quad |d| \leq 2\sigma \ then\ e = 20\sigma \end{cases}
\end{cases} \qquad (6)
$$

A non-Gaussian error is used to impose a weak constraint on the model at depth (Fig. 9). That
way, we correct the model drift without constraining a slow moderate variability or trend.
Basically, the hypothesis is that small to medium departures from the climatology ($2\sigma$ or less)
has an even probability. For instance, a 0.2 °C model warming at 2000 m due to a positive
North Atlantic Oscillation pattern must not be corrected as zero. Indeed, a 0.2 °C cooling is as
likely as the warming, since the climatology is the time average of those anomalies. So, only
large departures from climatology ($3\sigma$ or more) should be corrected. It corresponds to highly
unlikely events that are typical of model drifts. An interesting point is that model drift is often
corrected locally, downstream the outflow, before it spreads out (see Fig. 10). Ideally, it gives
a little regional correction instead of a large basin scale bias.
To validate this kind of assimilation, two global ocean simulations of several years, using
only the 3D-VAR large-scale biases correction in temperature and salinity, have been
performed. Due to the high computational cost of the system PSY4V3, the assimilation of
WOA13v2 below 2000 m has been tested with a global intermediate-resolution system at ¼°,
which is, in all other aspects, very close to the high resolution system PSY4V3. All in situ
observations have been used as well.
In practice, the assimilation of WOA13v2 climatological profiles below 2000 m in the system
concerns mostly some regions where the steep bathymetry might be an issue for the model
(Kerguelen Plateau, Zapiola Ridge, and Atlantic ridge). Figure 10 shows mean temperature
(left) and salinity (right) innovations (WOA13v2 climatological profiles minus model) in
2013 at 2865 m. The assimilation of these climatological profiles occurs more or less at the
same locations over the time period 2007-2016. Since the conditions of the system of
equations (6) relate to the density innovation, we have a perfect symmetry of the temperature
and salinity data which are assimilated. This has the effect of not disturbing the density
gradients too much.
If we focus on latitudes between 30° S and 60° S, Fig. 11 represents temperature (top panels)
and salinity (low panels) annual anomalies over depth (500 - 5000 m) and time (2007-2014).
The simulation on the left does not assimilate climatological vertical profiles while the



simulation on the right assimilates some. These maps demonstrate that the assimilation of
WOA13v2 below 2000 m is beneficial, reducing drifts below 2000 m. In the Antarctic
Circumpolar Current (ACC), the assimilation of these profiles makes it possible to maintain,
for instance, the Antarctic Bottom Water (see Gasparin et al., 2018 - Submitted in Journal of
Marine Systems). This also impacts the vertical repartition of the steric height, without
degrading the quality of the results comparing with profiles from the Argo network.

### 3.4    Construction of the forecast error covariance

The seasonally varying forecast error covariance is based on the statistics of a collection of
3D ocean state anomalies. This approach is based on the concept of statistical ensembles in
which an ensemble of anomalies is representative of the error covariance. In this way,
truncation no longer occurs and all that is needed is to generate the appropriate number of
anomalies. The way in which these anomalies are computed from a long numerical
experiment is described in Lellouche et al. (2013). In the previous system PSY4V2, a free
simulation was used to calculate the anomalies. For the system PSY4V3, the anomalies are
computed from a simulation in which only a 3D-VAR large scale bias correction of T/S has
been performed. In the following section, we evaluate the potential added value of this choice
on the quality of the analysis increments.

#### 3.4.1    Choice of the simulation from which to calculate the anomalies

The system PSY4V3 was run over the October 2006 – October 2016 period to catch-up the
real-time ("OPER" simulation), starting from 3D temperature and salinity initial conditions
based on the EN4 climatology. This simulation benefited from the full data assimilation
system, including the 3D-VAR biases correction and the SAM filter. Two other simulations
over the same period have been performed. The first one is a "FREE" simulation (without any
data assimilation) and the second one has exactly the same model tunings but only benefits
from the temperature and salinity 3D-VAR large-scale biases correction ("BIAS" simulation).
Figure 12 and Figure 13 show comparisons between this triplet of PSY4 simulations and two
observational products. The first product is the CMEMS/DUACS (Data Unification and
Altimeter Combination System) Merged-Gridded Sea Level Anomalies heights in delayed
time on a ¼° regular horizontal grid with a 1-day temporal resolution (Pujol et al., 2016). The
second one is the Roemmich-Gilson Argo monthly climatology on a 1° regular horizontal grid
(Roemmich and Gilson, 2009) which is commonly used in the oceanographic community.





Figure 12a,b,c shows the 2007-2015 SSH variability for the three simulations. SSH variability
difference is defined as the difference of SSH standard deviations from PSY4 simulations and
the DUACS product (Fig. 12d,e,f). Comparing to the variability of the DUACS product, the
fronts in high mesoscale variability regions such as the Gulf Stream, the Kuroshio, the
Agulhas current or the Zapiola eddy are misplaced in the FREE simulation. In the BIAS
simulation, these fronts are better positioned due to the large-scale correction of temperature
and salinity. However, this simulation presents more energy compared to DUACS, apart of
the main fronts. This corresponds to a leakage of vorticity from the fronts due to the mean
advection. Note that the gridded DUACS product also underestimates the variability as
wavelengths smaller than 200 km are barely resolved in the gridded fields. The mesoscale
features are well constrained in the OPER simulation with the information coming from
satellite data.
Time-averaged density differences along the equatorial Pacific between two ENSO events
("Oct-Dec 2008 minus Oct-Dec 2009"), computed from the PSY4 simulations and from the
Roemmich-Gilson Argo monthly Climatology, are shown in Fig. 13. The SCRIPPS Argo
product presents a higher density difference in the eastern part of the equatorial Pacific. It
corresponds to the change from moderate La Niña conditions early 2008 to moderate El Niño
conditions in 2009. The FREE simulation is not dense enough in the east compared to
observations particularly at the pycnocline depth ($1025 \ kg/m^3$ isopycn). The BIAS simulation
intensifies the density difference. The OPER simulation gets even closer to the SCRIPPS
Argo product. There is also an upward tilt of the density difference maximum in agreement
with the observations.
In summary, the BIAS simulation better represents the density fronts on the horizontal (Gulf
Stream) and on the vertical (Pacific pycnocline). The covariance matrix deduced from this
simulation has information on the density gradients that is well placed. This is valuable off the
equator though geostrophy, and at the equator to control the zonal pressure gradient. The
variance in sea level is stronger than the DUACS one (see Fig. 12e) but the most important
point for the construction of the anomalies is to have well-placed density gradients. In the
OPER simulation and as mentioned in Lellouche et al. (2013) in the description of the data
assimilation system SAM, an adaptive scheme will correct the variance and will give an
optimal model error variance based on a statistical test formulated by Talagrand (1998).





**3.4.2 Anomaly filtering**
The signal at a few horizontal grid "Δx" intervals in the model outputs on the native full grid
is not physical but only numerical (Grasso, 2000) and should not be taken into account when
updating an analysis. This is why several passes of a Shapiro filter have to be applied at the
anomalies computation stage in order to remove the very short scales that in practice
correspond to numerical noise. This can also help to filter out the noise from the covariance
matrix due to the sampling error (Raynaud et al., 2009).
To illustrate the impact of the anomaly filtering, we set up some experiments consisting in the
assimilation of a single altimeter track with different levels of filtering. These experiments
have been performed with a Mercator Ocean regional system at 1/12° using the SAM data
assimilation scheme, in order to reduce the high computing cost of the system PSY4V3 as
well as the time consuming to build different sets of anomalies at the global scale. Figure 14
shows SLA increments obtained with these different levels of anomaly filtering. It should be
noted that the anomaly filtering has a direct effect on the analysis increment, since the latter is
a linear combination of the anomalies.
Figure 14a represents SLA innovation along the single assimilated track. Figure 14b,c,d
represents the SLA increments obtained respectively with 10, 100 and 300 Shapiro passes as
the anomaly filtering mentioned above (corresponding approximately to a 3, 10 and 15
horizontal grid "Δx" intervals filter). We can see that the correction on the track remains more
or less the same. The strongest differences occur outside the track where the innovation
information is extrapolated.
Other experiments, closer to real time integration set up have been performed, assimilating all
the altimeter tracks available on a 7-day assimilation window, instead of one single track.
Figure 15 shows the SLA increments difference using 10 and 300 Shapiro passes as anomaly
filtering. The conclusions are the same as those concerning the experiments with a single
assimilated track. The correction on the tracks remains almost the same for the two levels of
filtering as small differences appear along the tracks. The strongest differences occur outside
the tracks where the innovation information is extrapolated to fill the gaps. Unfiltered
increments have small-scale structures that are statistical artifacts. Small structures can
cascade in the model, and stay trapped between the repetitive tracks, without correction by the
assimilation. This happens less when the filtering is performed on the anomalies beyond the
effective resolution of the model.

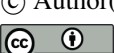



**3.5  Adaptive tuning of observation errors**
In order to refine the prescription of observation errors (instrumental and representativity
errors), adaptive tuning of observation errors for the SLA and SST has been implemented in
PSY4V3. The method has not been used for temperature and salinity vertical profiles because
of the reduced number of in situ data compared with satellite data. Then, 3D fixed observation
errors are used for the assimilation of in situ temperature and salinity vertical profiles. The
method consists in the computation of a ratio, which is a function of observation errors,
innovations and residuals (Desroziers et al., 2005). It helps correcting inconsistencies on the
specified observation errors. This ratio can be expressed as:

$$ratio = \frac{residual\ (innovation)^T}{observation\ error} \qquad (7)$$

Ideally, $ratio$ is equal to 1. When the ratio is less (respectively larger) than 1, it means that
the observation error is overestimated (respectively underestimated). The objective of this
diagnostic is to improve the error specification by tuning an adaptive weight coefficient acting
on the error of each assimilated observation. As a first guess of the method, the initial
prescribed observation error matches the one used in the previous system (Lellouche at al.,
2013) where the observation error variance was increased near the coast and on the shelves
for the assimilation of SLA, and increased only near the coast (within 50 km of the coast) for
the assimilation of SST.
Figure 16 represents the temporal evolution of the ratio defined in Eq. (7) for Envisat satellite.
At the beginning of the simulation, the observation error is overestimated (ratio less than 1).
The ratio tends to 1 after only a few weeks of simulation.
For SLA (Fig. 17), the a priori prescribed observation error is globally significantly reduced.
The median value of the error changed from 5 cm to 2.5 cm in a few assimilation cycles and
allows for better results. This method allows us to have more realistic and evolutive
observation error maps which can provide valuable information for the space agencies.
The realism of tropical oceans is crucial for seasonal forecasting applications. Tropical
Instability Waves (TIWs) can be diagnosed from SST (Chelton et al., 2000). These Kelvin
Helmholtz waves initiate at the interface between areas of warm and cold sea surface



temperatures near the Equator and form a regular pattern of westward-propagating waves.
Figure 18 gives an example of adjustment of the observation error to the model physics and
atmospheric variability. The SST anomalies in the equatorial Pacific clearly show the
propagation westwards of TIWs in the second half of the year. This is more pronounced
during episodes of La Niña (mid-2007 and mid-2010). The observation error anomalies
estimated by "Desroziers method" show that the error increases when these TIWs are more
marked. This can be explained by uncertainties in SST observations (clouds) and model shift
of the TIWs structures. The error decreases in the reverse case.
We have also performed an Empirical Orthogonal Function (EOF) analysis to assess the
variability of the SST observation error (Fig. 19). Mode 1 is associated to the seasonal cycle
and mode 2 (not shown) corresponds to the migration of the seasonal signal. Mode 3 is
associated to the inter-annual signal with for instance the transition La Niña / El Niño,
showing that the SST error is able to adapt both to the seasonal and inter-annual fluctuations.

## 15   4   Scientific assessment

This section describes the PSY4V3 system's quality assessment with diagnostics over
particular years, together with time series over multiyear periods. To evaluate the quality of
the system, the departure from the assimilated observations (SST, SLA, T/S vertical profiles
and sea ice concentration) is measured. Moreover, the analyses are also compared with
observations that have not been assimilated by the system such as tide gauges, velocity
measurements from drifting buoys, NOAA SST and AMSR sea ice concentration. NOAA
SST and AMSR sea ice analyses are not fully independent, since the upstream observations
are the same than for assimilated CMEMS OSTIA SST and OSI Sea Ice concentrations, but
comparisons to a variety of estimates using different algorithms and protocols provides a
useful consistency analysis.

### 26   4.1   SST

#### 27   4.1.1   Assimilated SST

OSTIA product is assimilated in the system PSY4V3. Compared to the previous system
PSY4V2, some large scale cold biases with respect to OSTIA are reduced in the Indian,
Eastern South Pacific, and western North Pacific (not shown). On the other hand, warm biases
are not reduced, especially in regions of strong inter-annual warm events such as the Eastern



Tropical Pacific where strong El Niño took place in 2015/2016, but also in the ACC, the Gulf
Stream and the Greenland Current (Fig. 20a). Some inconsistencies can be found between
OSTIA SST and in situ near surface temperature, particularly in the North Pacific where the
system PSY4V3 presents a cold bias compared to in situ near surface temperature but a warm
bias compared to OSTIA SST (Fig. 20b). Figure 20c shows the difference between drifting
buoys SST and the system PSY4V3 over the year 2015. The drifting buoys SST data are
present in the CMEMS in situ database used by Mercator but they have not been assimilated
in the system because the depth of these data is a nominal value and we chose to assimilate
only data with a measured depth value. Although we plan to assimilate these data in the future
system, we use currently this data as independent information. This allows us to see that SST
from in situ vertical profiles and SST from drifting buoys are coherent with each other. We
thus find again the cold bias highlighted by the comparison with  SST from in situ vertical
profiles in the North Pacific.
We checked also the time series of the mean and the RMS of the misfit (innovation) between
the observed SSTs and the model. For OSTIA SST, we obtain a mean warm bias of -0.1 °C
and a RMS error of 0.45 °C (Fig. 21). Seasonal fluctuations of the SST biases on global
average can be seen as a lack of stratification in the model, which causes stronger mid-latitude
warm biases during (boreal) summer (and a warm bias between 50 m and 100 m). For in situ
SST, the bias is smaller, suggesting that OSTIA might be colder than in situ near surface
observations on global average. We can notice a drop in the RMS of in situ surface data in
January 2014, which is due to the use of near real time observations, where most of the
surface observations do not have sufficient quality flag.
### 4.1.2   Comparison with an independent SST product
CLS (Collecte Localisation satellites) operates since 2002 a near real time oceanography data
service named CATSAT, for scientific, institutional or private users (support to fishery
management or to the offshore oil and gas industry). These data include satellite observations
as chlorophyll-a, SST and altimetry. Maps of SST are computed from Aqua/MODIS, S-
NPP/VIIRS and Metop/AVHRR infra-red sensors at 2 km resolution, using nighttime data
only to avoid diurnal warming effects. We can then evaluate the system ability to produce the
mesoscale by comparing with the CATSAT daily SST product. On Fig. 22, the CATSAT
daily snapshot can be considered as an independent dataset since the OSTIA SST assimilated
in the system has mostly seen microwave measurements during two weeks, as it was very


cloudy in the Gulf of Mexico. 31$^{st}$ of March 2016 is the first clear day showing well, from
infrared measurements, the Loop Current and other structures in the western part of the Gulf
of Mexico. The Loop Current is almost forming a closed meander. This is reproduced by the
system PSY4V3, as well as secondary structures like the filament in the North (Fig. 22).
Visible limitations of this 1/12° system concern the fine sub-mesoscale that can not be
resolved, and the lack of tidal mixing along Yucatan coasts (Kjerfve, 1981).
## 4.2   Temperature and salinity vertical profiles
For the T/S vertical profiles, we checked time series of the RMS of the difference between the
model analysis and the observations, for temperature on the left and for salinity on the right
(Fig. 23) in the whole water column. We compare observation and − climatology (red line),
previous system PSY4V2 (blue line), new system PSY4V3 (black line).
On global average, and compared to the previous system PSY4V2, the system PSY4V3
slightly degrades the temperature statistics (-0.03 °C) but it significantly improves the salinity
statistics by decreasing the 0-5000 m RMS salinity by 0.1 psu. This allows a more accurate
description of the water masses. This better balance arises from the new in situ errors that give
more weight to the salinity data (not shown). We can also notice that the systems are always
better than the climatology. The comparison to climatology is a minimum performance
indicator that the system must achieve. The differences with the climatology are worst from
the beginning of the year 2013. It can be explained by the fact that six different decades of
WOA13v2 monthly climatology can be found on the NODC website. We chose the available
2005-2012 decade (near of our time period simulation). So, in situ temperature and salinity
vertical profiles we assimilated in the system are coherent with this WOA13v2 product until
the end of year 2012 and this is no longer the case after.
Moreover, the system PSY4V3 experiences a slight warm bias (negative observation minus
forecast difference) in subsurface (25 - 500 m) on global average (not shown). For the year
2015, part of this signal comes from the strong inter-annual ENSO signals in the Tropical
Pacific where the near surface bias is also warm, as well as in the ACC and the Gulfstream.
Seasonal cold surface biases appear in the mid latitudes, linked with a lack of stratification
during summer. Summer warming is injected too deep and results in subsurface spurious
warming and too shallow mixed layer. However, these biases remain small on global average.



### 4.3   Sea Level

#### 4.3.1   Assimilated SLA

The system PSY4V3 is closer to altimetric observations than the previous one with a global forecast RMS difference of around 6 cm instead of 7 cm for the system PSY4V2 (not shown). This RMS difference is consistent with the observations errors (about 2 cm for altimeters and 4 cm for MDT). The statistics come from the data assimilation innovations computed from the forecast used as the background model trajectory, and give an estimate of the skill of the optimal model forecast. These scores are averaged over all seven days of the data assimilation window, which means the results are indicative of the average performance over the seven days, with a lead time equal to 3.5 days.

More precisely, on the year 2015, the SLA mean and RMS errors are considerably reduced in the new system PSY4V3 compared to the previous one (Fig. 24). The mean bias is reduced by 0.3 cm (from -0.8 cm to 0.5 cm) and the RMS is reduced by 2.4 cm (from 7.9 cm to 5.5 cm). This is mainly due to the use of the "Desroziers" method to adapt the observations errors online, which yields to more information from the observations being used (see Sect. 3.5). These improvements occur in nearly all regions of the ocean but are more pronounced in some regions (e.g. North Atlantic, Hudson Bay, Labrador Sea). In some others regions (e.g. Indonesian or west tropical Pacific), it remains some errors in sea level linked to the uncertainty in the MDT or missing parametrisations in the model (interaction wave-current, tides).

#### 4.3.2   Comparison to tide gauge data

The system PSY4V3 produces hourly outputs at the surface that can be compared with tide gauge measurements. For that, we used the BADOMAR product which is a specific processed tide gauges database developed and maintained at CLS and consists of filtered tide gauge data from the GLOSS/CLIVAR "fast" sea level data tide gauge network. These tide gauge data are corrected from inverse barometer effect and tides. High frequency model SSH compares well with tide gauges in many places, with a slight improvement in PSY4V3 with respect to PSY4V2 (not shown). The best agreement between the system PSY4V3 and tide gauges is found in the tropical band, as can be seen in Fig. 25, while shelf regions and closed seas are less accurate. This confirms the latitude dependence of the correlation between tide gauges and satellite altimetry or modelled SSH discussed in Vinogradov and Ponte (2011) or Williams and Hugues (2013).





The improvements related to water masses and SLA lead to a correct Global Mean Sea Level
(GMSL) trend. We checked the system GMSL by comparing the results with recent estimated
trend from the paper of Chambers et al. (2017). We found for the model a trend of 3.2 mm yr⁻
$^1$ over the PSY4V3 simulation time period which is coherent with DUACS value (3.17 ± 0.67
mm yr$^{-1}$). Moreover, the temporal evolution of the global mean model SSH is coherent and
phased with the observations.

### 4.4 Sea ice concentration

#### 4.4.1 Assimilated sea ice concentration

The system PSY4V3 assimilates OSI SAF sea ice concentration in both hemispheres with a
monovariate/monodata scheme. As expected, PSY4V3 is closer to the observations than the
previous system PSY4V2 (not shown), in which no sea ice observations had been assimilated.
As illustrated by Fig. 26, the system PSY4V3 has a slight overestimation of ice during the
melting season in summer (up to 3 % on average in both hemispheres). Conversely, the mean
error is stronger on average during winter (10 to 20 % underestimation, depending on the
year). RMS errors are also larger during summer (up to 20 % in the Arctic and 30 % in the
Antarctic with respect to OSI SAF observations), and drop to less than 10 % in winter. These
RMS errors quantify the capacity of the system to capture weekly time changes in the ice
cover.
We have also checked the evolution of the sea ice volume diagnosed by the system PSY4V3
which assimilates observations of sea ice concentration with a monovariate/monodata scheme.
The data assimilation scheme SAM produces increment of sea ice concentration which is the
unique sea ice correction applied in the model using the Incremental Analysis Update (IAU)
method described in Lellouche et al. (2013). The sea ice volume then adjusts to this correction
considering a constant sea ice thickness. No sea ice thickness observations are assimilated in
the system. The risk is therefore to obtain unrealistic drifts or trends of the unconstrained sea
ice volume. Presently, sea ice volume retrievals from satellites are associated with large
uncertainties (Zygmuntowska et al., 2014). Consequently, modelled sea ice volume is difficult
to validate and one of the solution is to compare modelled sea ice volume from several
systems.
Figure 27 shows the 2007-2016 evolution of sea ice volume for the system PSY4V3, the
PIOMAS modelled product (Schweiger et al., 2011) and the CMEMS GREP (Global




Reanalysis Ensemble Product, http://marine.copernicus.eu/documents/QUID/CMEMS-GLO-
QUID-001-026.pdf) composed by four global ¼° reanalyses and the ensemble mean with the
associated spread from the four members. All the modelled sea ice volumes present the same
2007-2016 inter-annual variability. PSY4V3 and PIOMAS are included in the spread whose
range is reduced over time from 4,000 km$^3$ in 2007 to 3,000 km$^3$ from the year 2012. The
GLORYS2V4 reanalysis is known to have a large sea ice volume compared to other
reanalyses (Chevallier et al., 2017). Although we use the same method for the assimilation of
sea ice concentration in GLORYS2V4 and PSY4V3, the sea ice volume diagnosed by
PSY4V3 lies in values ranging between 13,000 and 15,800 km$^3$, in a better accordance with
GREP and PIOMAS products.
**4.4.2   Contingency table analysis**
The contingency table analysis approach described in Smith et al. (2016) has been applied to
evaluate sea ice extent as compared to observation. Satellite ice concentration coming from
AMSR2 (L1B brightness with a NASA team 2 algorithm to compute sea ice concentration)
has been used as independent observation to provide a general assessment in the detection of
false alarms if ice coverage. Although this type of evaluation is usually done on forecasts, we
used hindcasts. For the computation of the statistics we have used a stereo-polar grid at a 20
km resolution. In each cell of that grid we have then computed binary values corresponding to
ice/open water conditions for the model and the sea ice observations by using a 40 %
concentration threshold. We have also restricted our study to the Proportion Correct Total
(PCT), following the conclusion of Smith et al. (2016), saying that it was more insightful to
refer to the PCT rather than others proportions. The PCT quantity is defined as PCT = (Hit ice
+ Hit water)/n (see Table 3), where n is the total number of observations with a sea ice
concentration greater than 15 %. A value of one corresponds to a perfect score.
Figure 28 shows times series of PCT for PSY4V2 and PSY4V3 systems. The lower PCT
values are due mostly to an excessive melt in spring and summer for both Arctic and
Antarctic. However, the assimilation of sea ice concentration improves significantly the total
hit rate during these periods.
**4.5   Currents**
The aim of this section is to use velocity observations which were not assimilated in the
system to assess the level of performance of PSY4V3 compared to the previous PSY4V2



system. The mean currents are checked by comparing the model to velocity observations
coming from Argo floats when they drift at the surface and in situ Atlantic Oceanographic and
Meteorological Laboratory (AOML) surface drifters. A paper by Grodsky et al. (2011)
revealed that an anomaly in the drogue loss detection system of the Surface Velocity Program
buoy had led to the presence of undetected undrogued data in the "drogued-only" dataset
distributed by the Surface Drifter Data Assembly Center. Rio (2012) applied a simple
procedure using altimeter and wind data to produce an updated dataset, including a drogue
presence flag as well as a wind slippage correction. We therefore used this new "drogued-
only" surface drifter dataset coming from CMEMS in situ TAC
(http://marine.copernicus.eu/documents/QUID/CMEMS-INS-QUID-013-044.pdf) to check
mean model currents.
Figure 29 represents zonal drift innovation for PSY4V2 and PSY4V3 systems. Although
some biases persist, mostly in the western tropical basins, significant improvements are
obtained almost everywhere with the new system PSY4V3, and more particularly in the
equatorial Pacific. The mean bias is reduced (from 0.1 m s$^{-1}$ to 0.08 m s$^{-1}$), the South
Equatorial Current is slower and there is also less noise in PSY4V3. Improvements are also
obtained, to a lesser extent, for meridional drift (not shown). The velocities have been slightly
improved in terms of velocity values but also in terms of currents direction (angle between
observed and modelled velocities). The mean angle difference is reduced from 9.1 degrees to
7.2 degrees. These improvements can be attributed to the new MDT used and the more
adapted filtering of anomalies. However, large biases persist in the western tropical Pacific
(very strong in 2015 because of the strong El Niño event) with a spurious extension of the
northern branch of the South Equatorial Current. This is probably linked to the uncertainty
still present in the MDT and unresolved or missed parameterized physical processes.
More locally, a comparison of the 2007-2015 averaged drifts from the system PSY4V3 and
the observations over the Indonesian region has been performed (not shown). Currents in this
region are very difficult to resolve because of the many narrow straits and the strong tidal
mixing. The retroflection of the westward South and North Equatorial Currents (along Papua
and near 12° N) into the eastward North Equatorial Counter Current (near 4° N) are well
reproduced structures in the Pacific. The system South Equatorial Current is a little too strong
at the edge of the warm pool but it is about the only weakness. The complex flow in the
Sulawesi Sea, the Makassar Strait and the South China Sea is well reproduced by the system.
The correlation is 0.70 (respectively 0.64) for the zonal (respectively meridional) velocity.



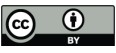

## 5    Summary and ways for improvement of the future system

The Mercator Ocean system PSY4V3, in an operational mode since October 19, 2016, benefits of many important updates. PSY4V3 has a quite good statistical behaviour with an accurate representation of the water masses, the surface fields and the mesoscale activity. Most of the components of the system PSY4V3 have been improved compared to the previous version: global mass balance, 3D water masses, sea level, sea ice and currents. Major variables like sea level and surface temperature are hard to distinguish from the data.

In this paper, the updates showing the highest impact on the products quality have been illustrated and evaluated separately. A particular focus was therefore made on the initialization, the correction of precipitation, the assimilation of climatological temperature and salinity in the deep ocean, the construction of the forecast error covariance and the adaptive tuning of observations error.

Initial climatological condition has been improved in order to be more consistent with the vertical profiles of temperature and salinity which has been assimilated thereafter. Rather than taking directly the climatological temperature and salinity of the month corresponding to the start of the simulation, we performed a pointwise linear regression, allowing to obtain an initial condition at the appropriate time and based only on real observations. One-year free simulations have been performed and show that biases are globally reduced.

Uncertainties inherent to atmospheric analyses and forecasts can induce large errors in the ocean surface fluxes. For instance a slight shift in the position of a storm can induce local errors in salinity, temperature and currents. In the tropical band, precipitations are systematically overestimated. Moreover, large scale salinity biases can appear because the global average freshwater budget is not closed. For this reason, IFS ECMWF atmospheric analysed and forecasted precipitations have been corrected at large scale using satellite-based PMWC product. This correction is beneficial in many areas, reducing the magnitude of the near-surface salinity fresh mean bias in the Tropics down to 0.5 psu.

Due to unresolved process and inaccurate atmospheric forcing, the model may also drift at depth. To keep some water mass properties, the DRAKKAR group used restoring of temperature and salinity toward annual climatology of Gouretski and Koltermann (2004) in specific areas. This choice was driven by the Antarctic Bottom Water restoring zone where this climatology is recognized as the more suitable. For Mercator systems which assimilate observations in a multivariate way, the problem can be more critical because of the





deficiencies of the background errors for extrapolated and/or poorly observed variables. To
overcome these deficiencies, vertical climatological T/S profiles have been assimilated below
2000 m using a non-Gaussian error at depth, allowing the system to capture a potential
climate drift in the deep ocean. In practice, the assimilation of climatological profiles below
2000 m in the system PSY4V3 concerns mostly some regions where the steep bathymetry
might be an issue for the model (Kerguelen Plateau, Zapiola Ridge, and Atlantic ridge). This
kind of assimilation reduces drifts below 2000 m and impacts the vertical repartition of the
steric height, without degrading the quality of the results comparing with the profiles from the
Argo network.
We have also proposed solutions to reduce some problems related to linearity and stationarity
hypotheses in the assimilation schemes. The first one concerns the construction of the forecast
error covariance. Rather than calculating the anomalies from a free simulation, we chose to
calculate them from a simulation benefiting only of the 3D-VAR large-scale biases correction
in temperature and salinity and representing better the density fronts on the horizontal and on
the vertical. Moreover, anomalies have been filtered in order to remove the scales beyond the
effective resolution of the model. The second one concerns the tuning of the observations
errors. Adaptive tuning of SLA and SST errors has been successfully implemented. It allows
us to have more realistic and evolutive SLA and SST error maps.
All these scientific and technical choices have been validated and integrated in the system
PSY4V3 which has been evaluated for the period 2007-2016 by means of a thorough
procedure involving statistics of model departures from observations. The system PSY4V3 is
close to SLA along track observations with a forecast (range 1 to 7 days) RMS difference
below 6 cm. Moreover, the correlation of the system PSY4V3 with tide gauges is significant
at all frequencies, however many high frequency fluctuations of the SSH might not be
captured by the system because tides or pressure effects are not yet included. The description
of the ocean water masses is very accurate on average and departures from in situ
observations rarely exceed 0.5 °C and 0.1 psu. In the thermocline, RMS errors reach 1 °C and
0.2 psu. In high variability regions like the Gulf Stream, the Agulhas Current or the Eastern
Tropical Pacific, RMS errors reach more than 2 °C and 0.5 psu locally. A warm bias persists
in subsurface, with peaks in high variability regions such as the Eastern Tropical Pacific, Gulf
Stream or Zapiola. Most departures from observed SST products do not exceed the intrinsic
error of these products (around 0.6 °C).

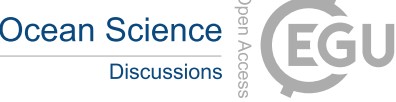



A global comparison with independent velocity measurements (surface drifters) shows that
the location of the main currents is very well represented, as well as their variability.
However, surface currents of the mid latitudes are underestimated on average. The
underestimation ranges from 20 % in strong currents to 60 % in weak currents. Some
equatorial currents are overestimated, and the western tropical Pacific still suffer from biases
in surface currents related to MDT biases. On the contrary the orientation of the current
vectors is better represented.
Lastly, the system reproduces the sea ice seasonal cycle in a realistic manner. However, the
sea ice concentrations are overestimated in the Arctic mainly during winter (due to
atmospheric forcing errors and too much sea ice accumulation) and in the Antarctic during
austral winter. They are underestimated during austral summer (too much sea ice melt and
errors caused by the rheology parameterization of the sea ice model). A contingency table
analysis approach has been also used to evaluate sea ice extent as compared to observations.
This approach shows clear improvements due to the assimilation of sea ice concentration in
the system PSY4V3.
Remarkable improvements have been achieved with the system PSY4V3 compared to the
previous version. However, some biases have been highlighted in the ocean surface features
as well as the 3D ocean structure at basin, sub-basin and local scales. The simulation biases
may be due to the initial state (especially in the deep layer where historical observation data
are rare), the atmospheric forcing uncertainties, the river runoff approximations, the efficiency
of the assimilation scheme, and the model errors induced by unresolved or parameterized
physical processes. Numerous projects have already been set up at Mercator Ocean to propose
innovative solutions. The integration of the ingredients from these projects into the future
CMEMS global high resolution system is planned for 2019. The improvement of numerical
simulations could thus be carried out, based on sensitivity tests on some model parameters
(e.g. coastal runoffs, atmospheric forcing, high frequency phenomena including tides, more
sophisticated sea ice model, interaction and retroaction between ocean currents and waves). It
is also planned to assimilate new types of observations in the system (drifting buoys SST,
higher resolution SST (L3 products), satellite sea surface salinity, velocity observations from
AOML surface drifters, and deep-ocean observations from Argo surface floats) to better
constrain the modeled variables and to overcome the deficiencies of the background errors in
particular for extrapolated and/or poorly observed variables. Another important issue is to use
a shorter assimilation time window and a 4D analysis in the assimilation scheme to better





correct the fast evolving processes. The next version of the global high resolution system will
also include seasonal errors for in situ vertical profiles already used in the CMEMS eddy-
resolving 1992-2016 reanalysis GLORYS at 1/12° horizontal resolution, which is based on
the system PSY4V3 and will appear on CMEMS catalogue in April 2018.

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



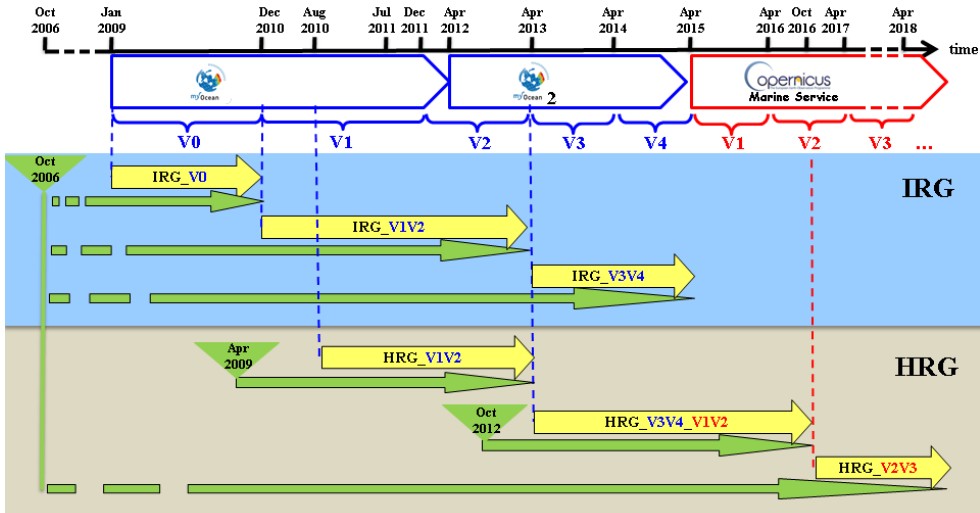

**Figure 1:** Timeline of the Mercator Ocean global analysis and forecasting systems for the various milestones (from V0 to V4) of past MyOcean project and for milestones V1, V2, V3 of the current CMEMS. Real-time productions are in yellow. Available Mercator Ocean simulations are in green including the catch-up to real-time. Global Intermediate Resolution (respectively High Resolution) system at 1/4° (respectively 1/12°) is referred to as IRG (respectively HRG). Milestones are written in blue for MyOcean project and in red for CMEMS.





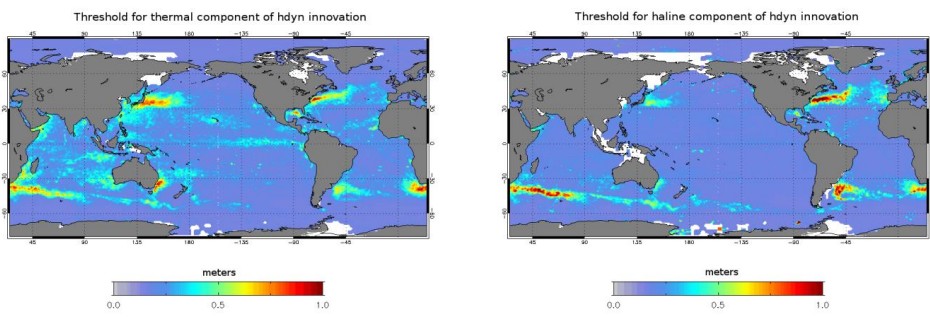

**Figure 2:** Thresholds used for QC2 for thermal component of dynamical height innovation (left panel: $threshold_T$) and for haline component of dynamical height innovation (right panel: $threshold_S$). Units are meters.





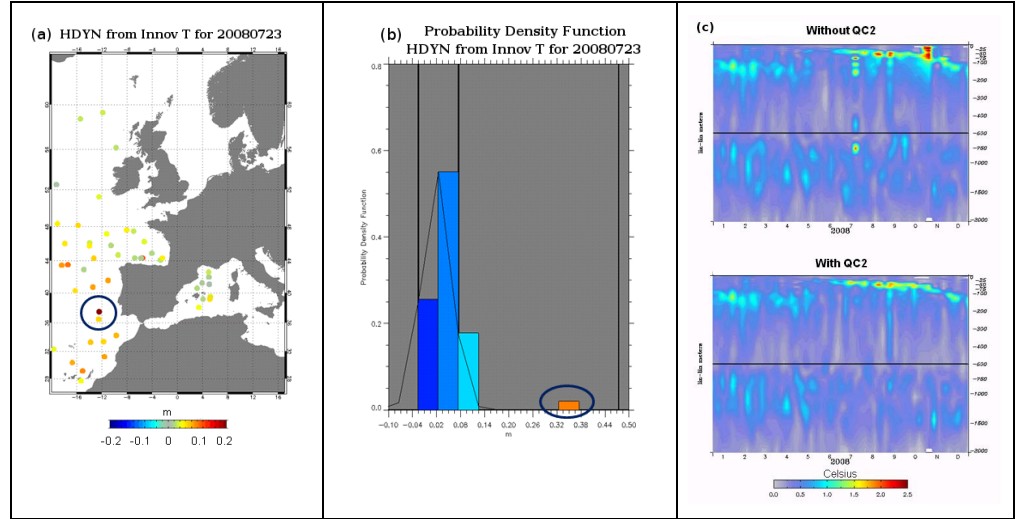

**Figure 3:** Statistics in the Azores region: a) absolute value of dynamical height innovations (in meters) from
temperature innovations for the 7-day assimilation cycle from 16 July 2008 to 23 July 2008, b) PDF of theses
dynamical height innovations (the value 0.3 m appears in the tail of the PDF), c) RMS innovation with respect to
the vertical temperature profiles over the year 2008 for two "twins" simulations (without and with QC2). Theses
last scores are averaged over all seven days of the data assimilation window, with a lead time equal to 3.5 days.
Units are °C.





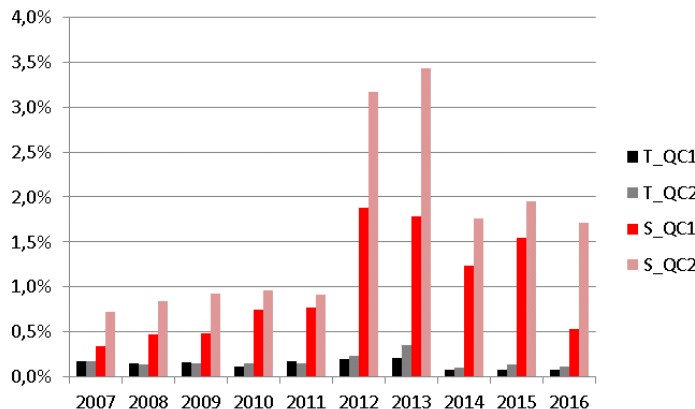

2   **Figure 4:** Statistics of suspicious temperature (T) and salinity (S) detected by QC1 (T_QC1 and S_QC1) and by
3   QC2 (T_QC2 and S_QC2) quality controls as a function of year in the PSY4V3 2007-2016 simulation time
4   period.



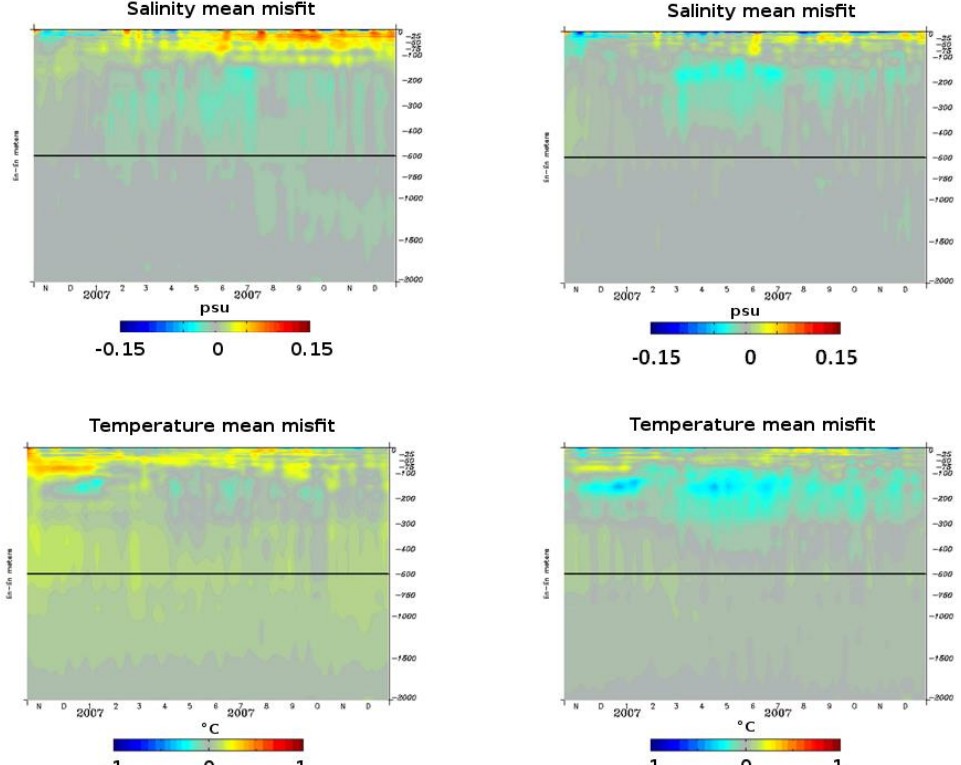

**Figure 5:** Diagnostics (time series) with respect to the vertical temperature and salinity profiles over the October
2006 - December 2007 period. Mean misfit between observations and model for salinity (top panels, units in
psu) and for temperature (low panels, units in °C), starting from WOA09 climatology (left panels) and robust
EN4 (right panels).





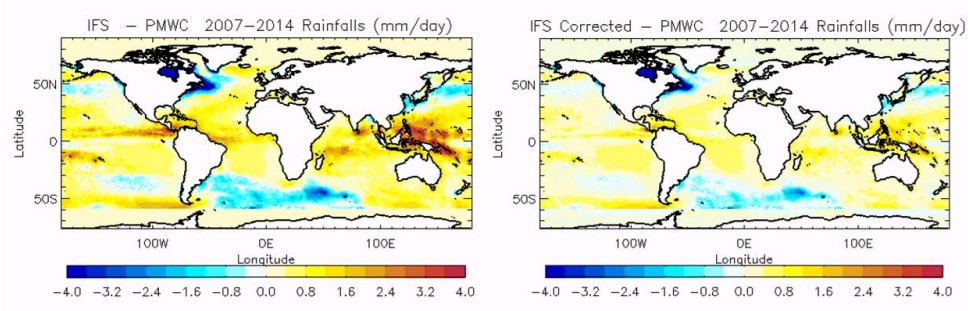

2 **Figure 6:** Mean 2007-2014 IFS ECMWF atmospheric precipitation bias (units in mm day$^{-1}$) with respect to
3 PMWC product without (left map) and with (right map) correction.





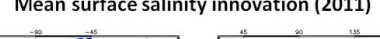

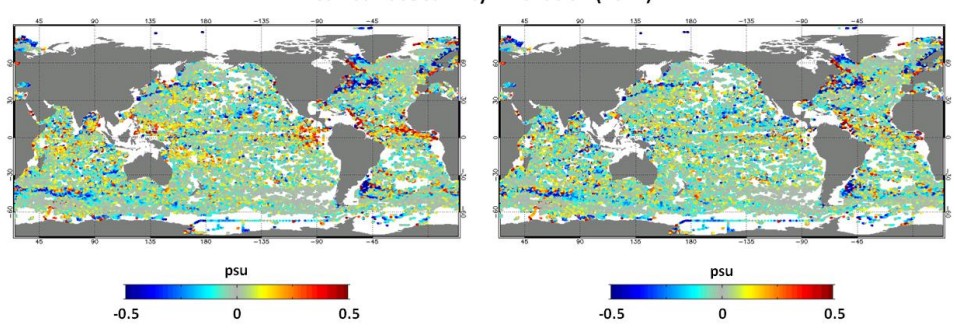

**Figure 7:** Mean surface salinity innovation (difference between the assimilated observation and the model, units
in psu) on the year 2011. On the left, the innovation resulting from the use of the original IFS field, and on the
right, the innovation resulting from the use of the corrected IFS field.



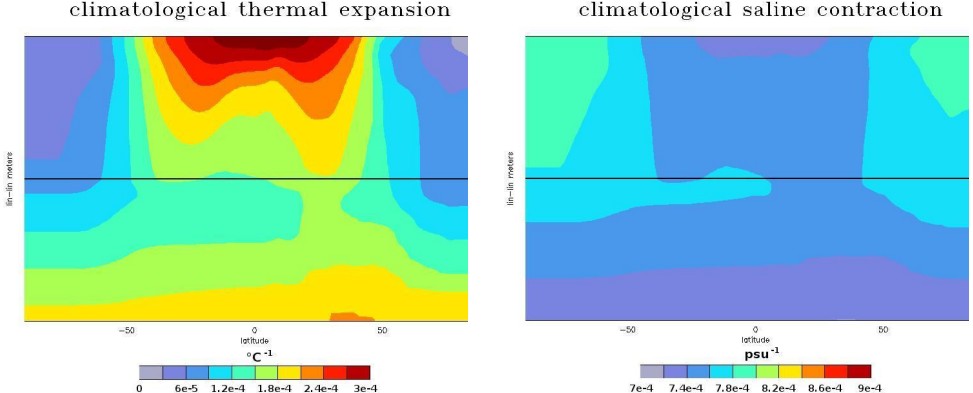

1  **Figure 8:** Climatological thermal expansion (°C⁻¹) and saline contraction (psu⁻¹) as a function of the latitude and
2  the depth.





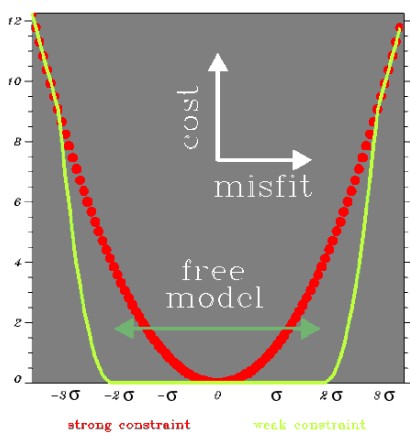

**Figure 9:** Non-Gaussian error for climatology (corresponding to a weak constrain of the system in green). A cost equal to zero corresponds to an infinite observation error, namely a system operation in a free mode (without assimilation of climatology).





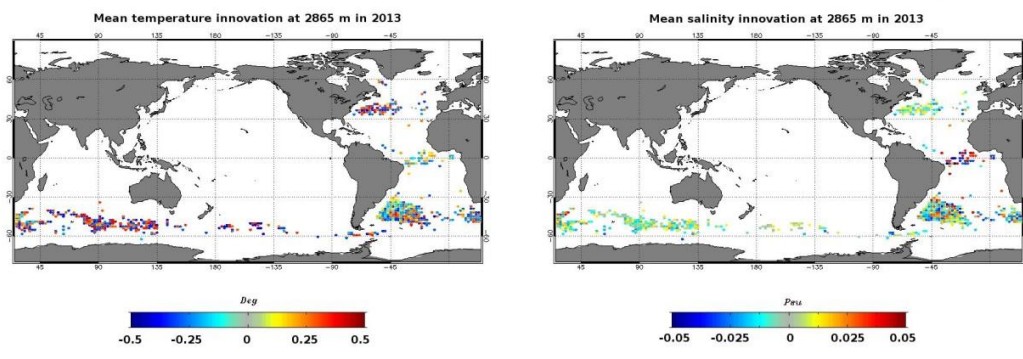

1    **Figure 10:** Mean temperature (on the left, units in °C) and salinity (on the right, units in psu) innovations in
2    2013 at 2865 m for the system PSY4V3.

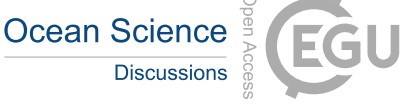

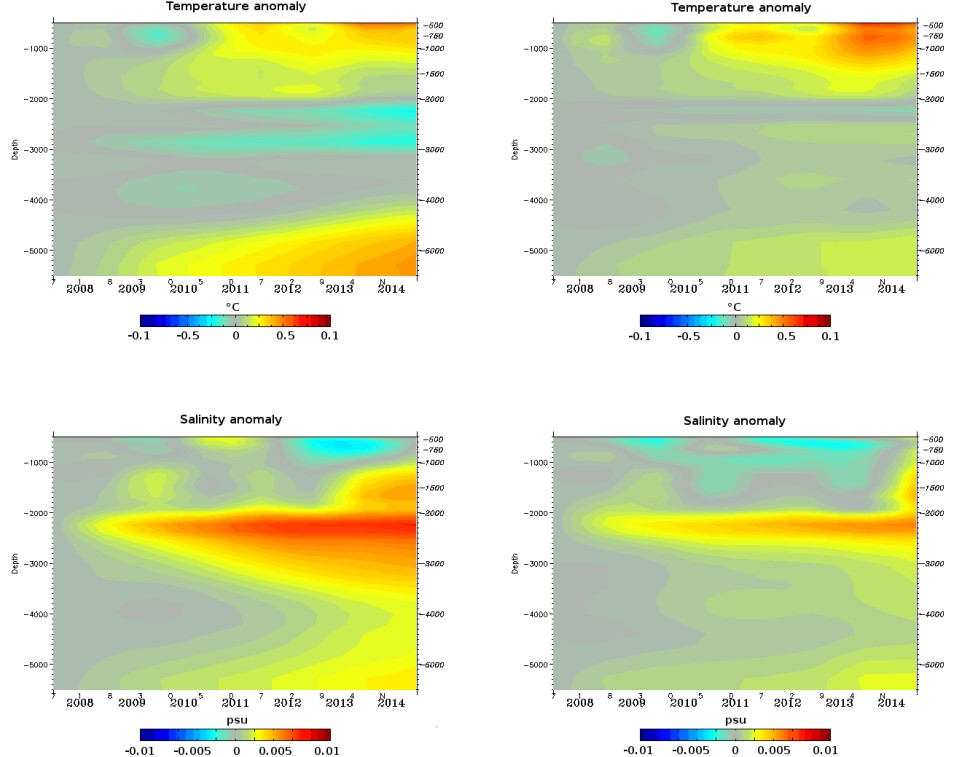

**Figure 11 :** Temperature (top panels, units in °C) and salinity (low panels, units in psu) annual anomalies over depth (500-5000m) and time (2007-2014) for latitudes between 30° S and 60° S. The simulation on the left does not assimilate climatological vertical profiles while the simulation on the right assimilates them. Annual anomaly for a specific year is computed as the difference between the annual mean of this year and the annual mean of the year 2007.



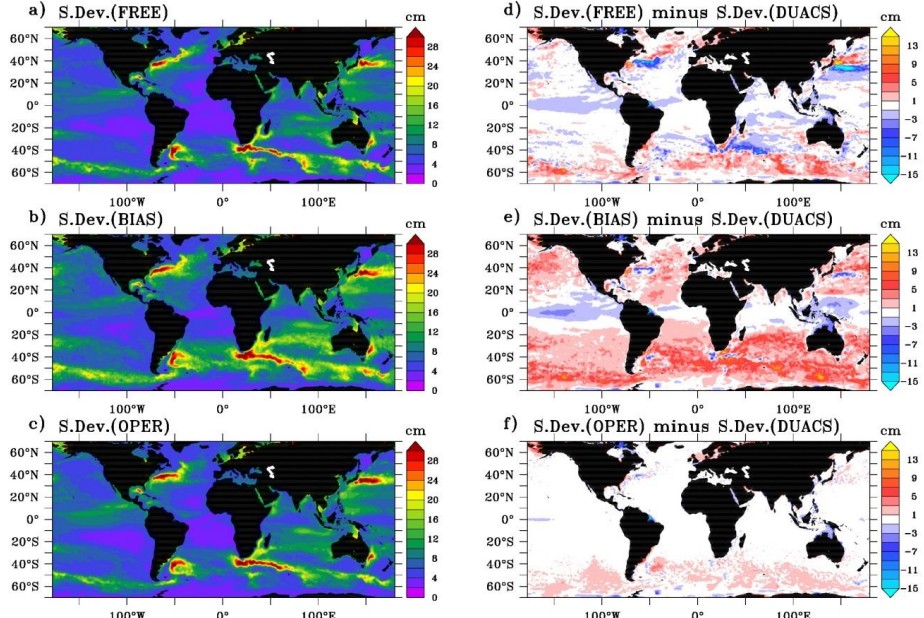

**Figure 12:** 2007-2015 SSH standard deviation (diagnostics made with 1 point every 3 horizontally and 1 day every 5) of the 1/12° PSY4 simulations (a,b,c) and difference of SSH model standard deviation with the one of DUACS product (d,e,f). Units are cm.




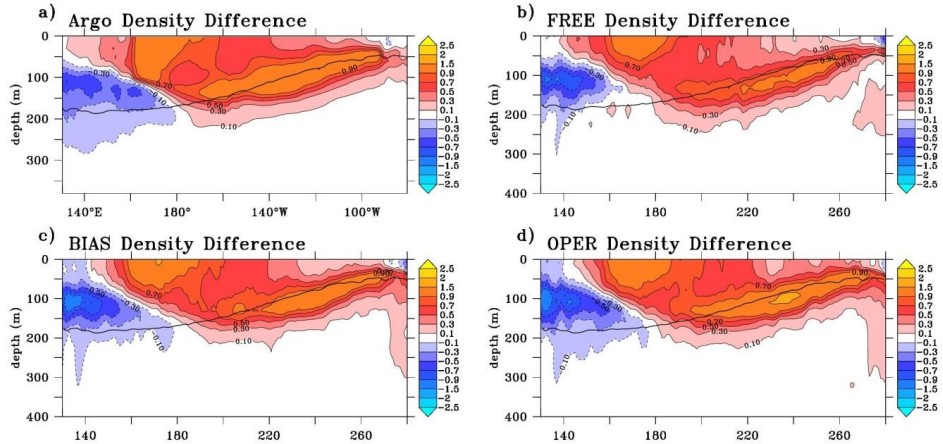

**Figure 13:** Density difference "OCT-DEC 2008 minus OCT-DEC 2009" in the equatorial Pacific (2° S-2° N)
above 400 m depth (a-d) from the SCRIPPS Argo product (a), and the three 1/12° PSY4 FREE, BIAS and OPER
simulations (b-d). The black line indicates the 2007-2015 Argo mean position of the pycnocline depth (isopycn
1025 kg m$^{-3}$).





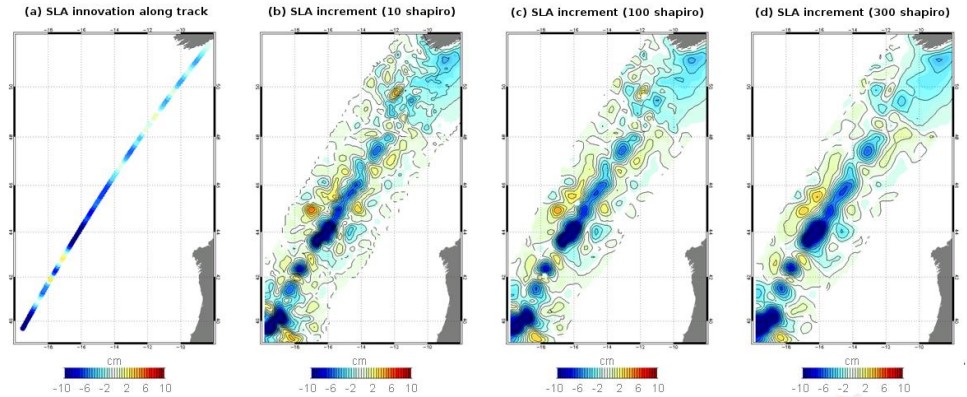

2 **Figure 14:** SLA innovation along a single assimilated track altimeter (a). SLA increments respectively with 10
3 (b), 100 (c) and 300 (d) Shapiro passes as anomaly filtering. These experiments have been performed with a
4 system at 1/12°. Unit is cm.



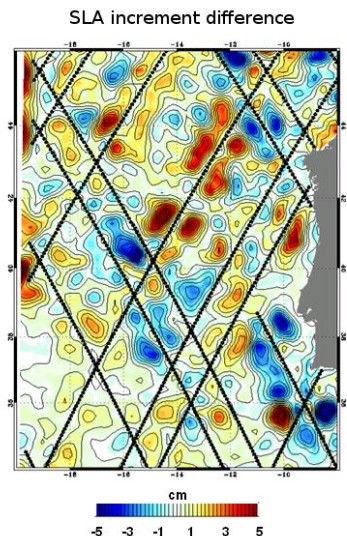

2 **Figure 15:** SLA increment difference using 10 and 300 Shapiro passes as anomaly filtering. The black lines
3 represent the position of the assimilated altimeter tracks. Unit is cm.





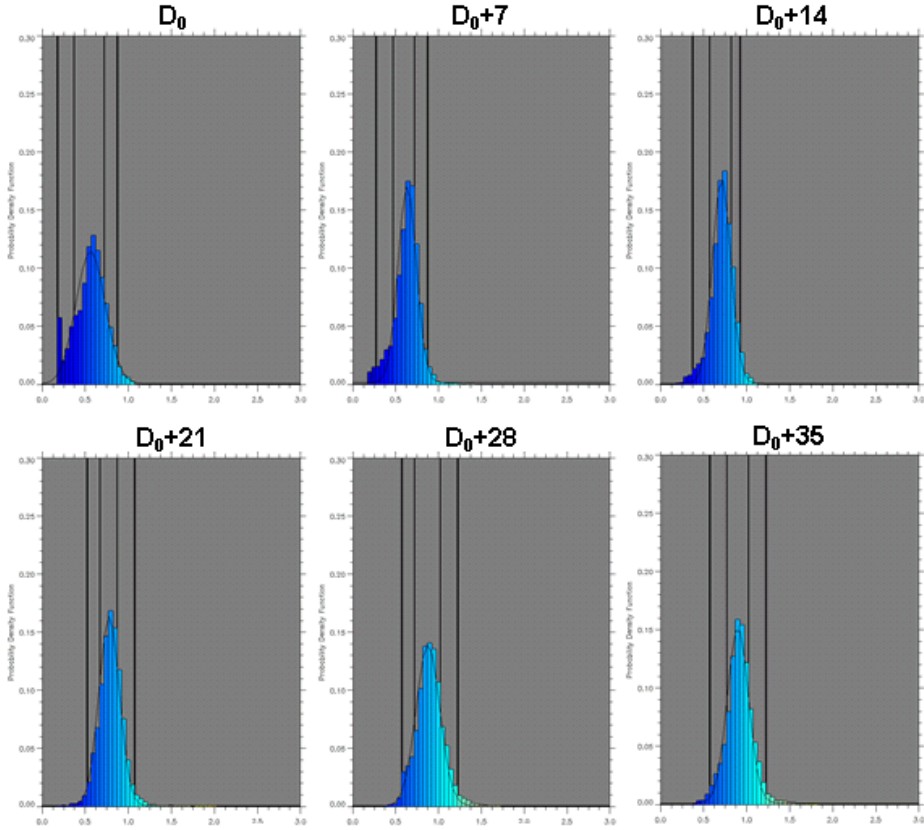

2 **Figure 16:** Evolution of the PDF of the ratio for Envisat satellite from $D_0$ to $D_0+35$ days. $D_0$ corresponds to the
3 first day where Envisat is assimilated by the system.



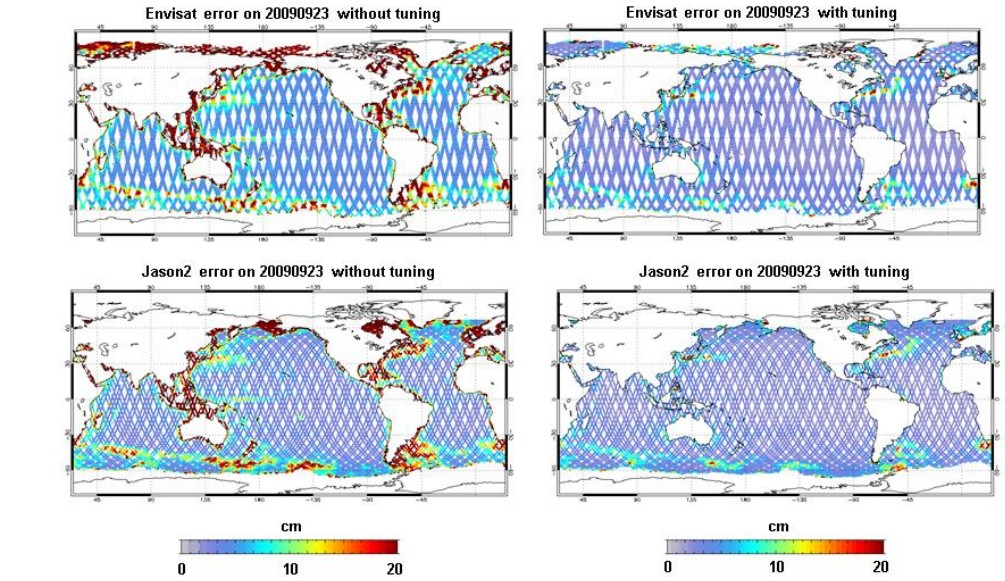

2  **Figure 17:** Envisat (top panels) and Jason2 (law panels) satellite observation errors used on the 7-day
3  assimilation cycle ending September, 23, 2009 without tuning (left panels) and with tuning (right panels)
4  method. Unit is cm.



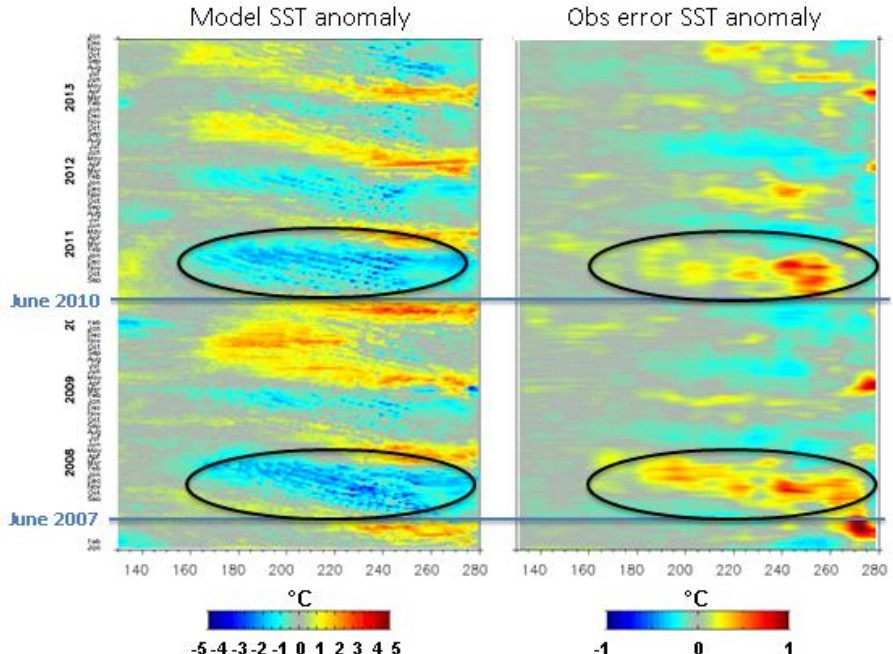

**Figure 18:** Evolution in time of model SST anomaly (on the left) and SST observation error anomaly tuned by
"Desroziers" method (on the right) for a section at 3° N. The blue lines represent the beginning of La Niña
episodes (mid-2007 and mid-2010). The black ellipses highlight periods when TIWs are more marked. Units are
°C.





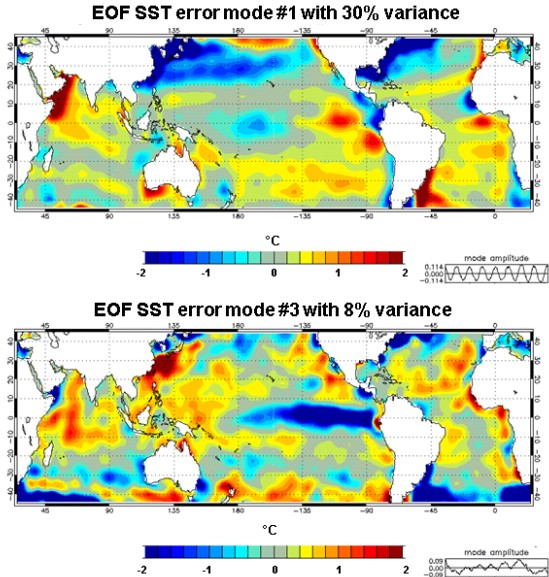

2 **Figure 19:** 1st EOF (top panel) and 3th EOF (bottom panel) of sea surface temperature observation error (°C)
3 over the 2007-2015 time period. The time series at the bottom of each panel correspond to the mode amplitude.

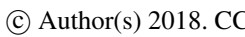



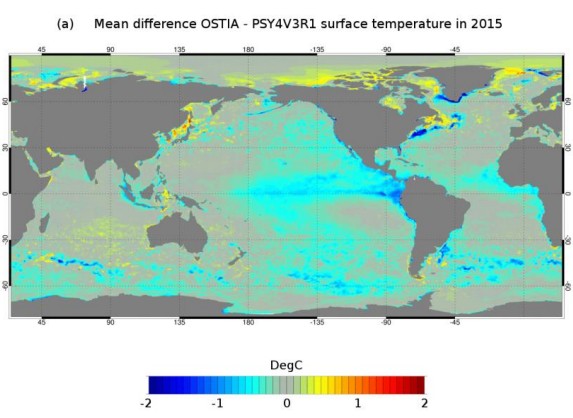

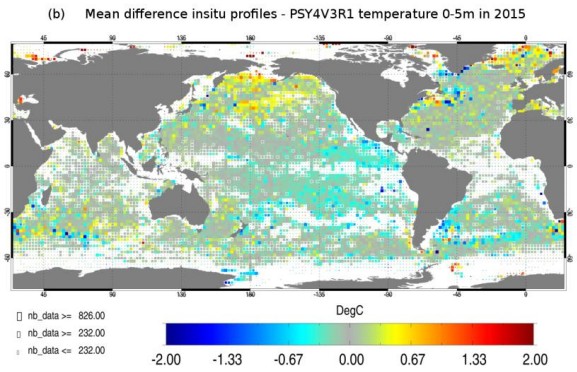

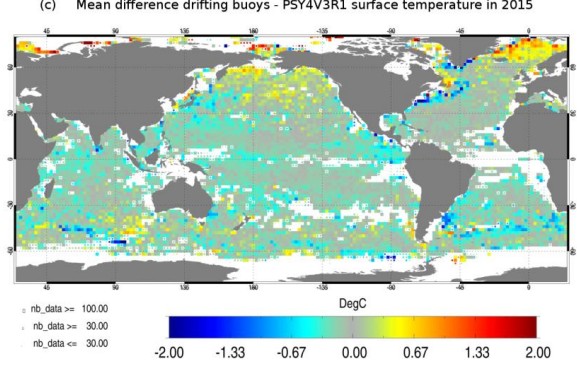

**Figure 20:** Mean SST residuals (units in °C) over the year 2015: OSTIA SST minus PSY4V3 (a), in situ SST
minus PSY4V3 (b) and drifting buoys SST minus PSY4V3 (c).



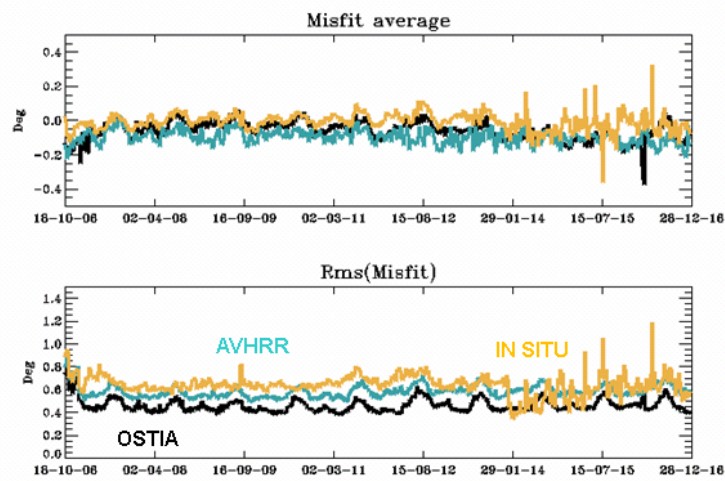

2 **Figure 21:** Time series of SST (units in °C) global misfit average (top) and RMS (bottom) for OSTIA
3 observations (black line, assimilated), NOAA AVHRR observations (blue line, not assimilated), and in situ
4 observations (orange line, assimilated), from October 2006 to December 2016.

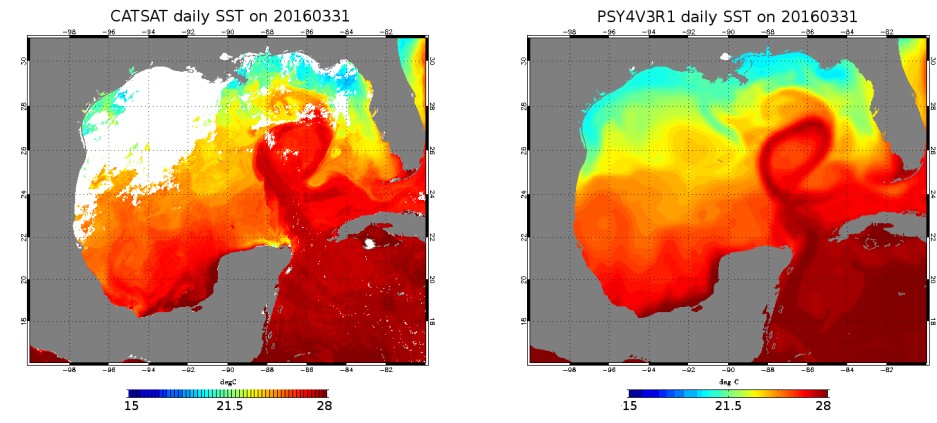

2  **Figure 22:** High resolution CATSAT SST from CLS (on the left) and PSY4V3 SST (on the right) on March 31,
3  2016. Unit is °C.





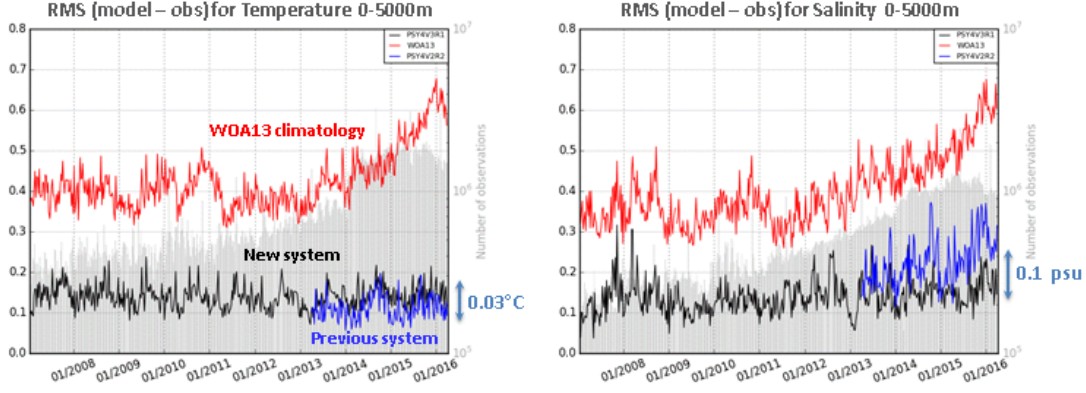

**Figure 23:** Time series of the 0-5000m RMS difference between the model analysis and the in situ observations
for previous system PSY4V2 (in blue), new system PSY4V3 (in black) and the WOA13v2 climatology (in red).
Left panel: temperature (unit in °C), right panel: salinity (unit in psu). Time series of the number of available
observations appear in grey.



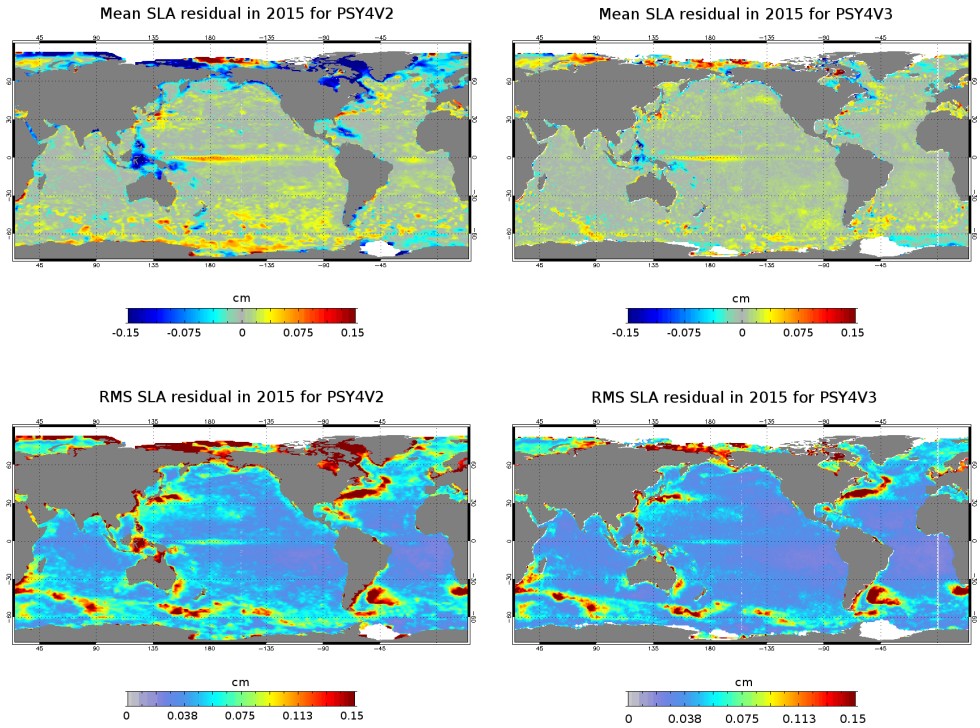

2 **Figure 24:** Mean residual errors (top panels) and RMS residual errors (low panels) of SLA in 2015, for the
3 previous system PSY4V2 (on the left) and the new system PSY4V3 (on the right). Unit is cm.





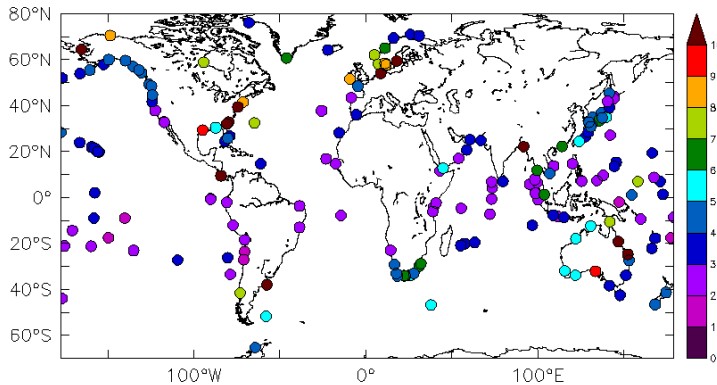

2  **Figure 25:** Sea surface height RMS difference between tide gauges observations and the system PSY4V3 for the
3  year 2015. Unit is cm.




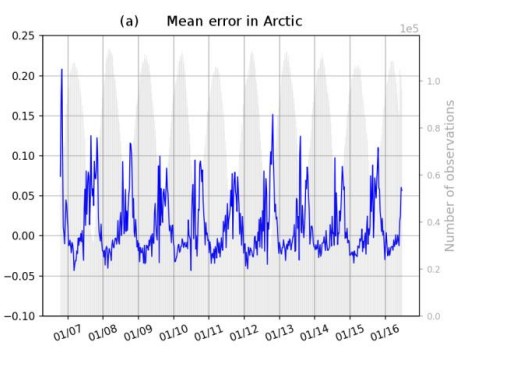
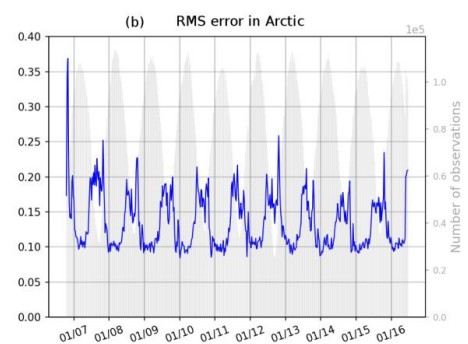

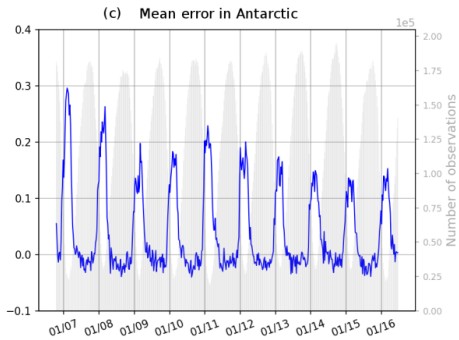
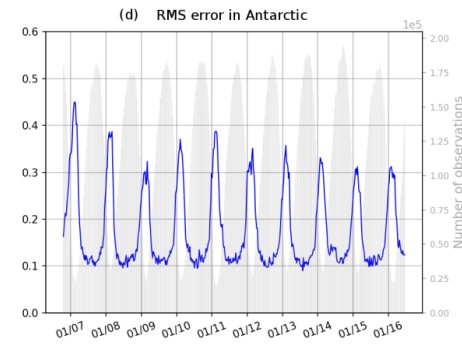

**Figure 26:** Time series of (observation-forecast) mean (a and c) and RMS (b and d) differences of sea ice concentration (0 means no ice, 1 means 100 % ice cover) in the Arctic Ocean (a and b) and Antarctic Ocean (c and d). The assimilated observations are the sea ice concentrations from OSI TAC. Time series of the number of available observations appear in grey.





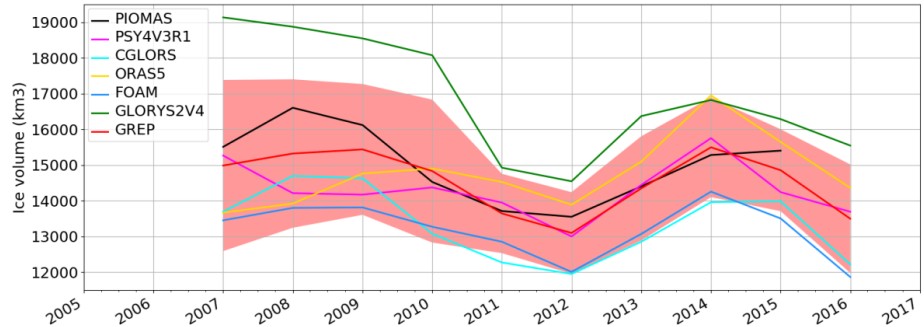

**Figure 27:** Time series over the 2007-2016 period of the sea ice volume in Arctic for several systems: GREP
composed by the four members GLORYS2V4 from Mercator Ocean (France), ORAS5 from ECMWF,
FOAM/GloSea from Met Office (UK) and C-GLORS from CMCC (Italy); PSY4V3 from Mercator Ocean
(France); PIOMAS product. The spread of GREP product is represented in light red. Unit is km$^3$.





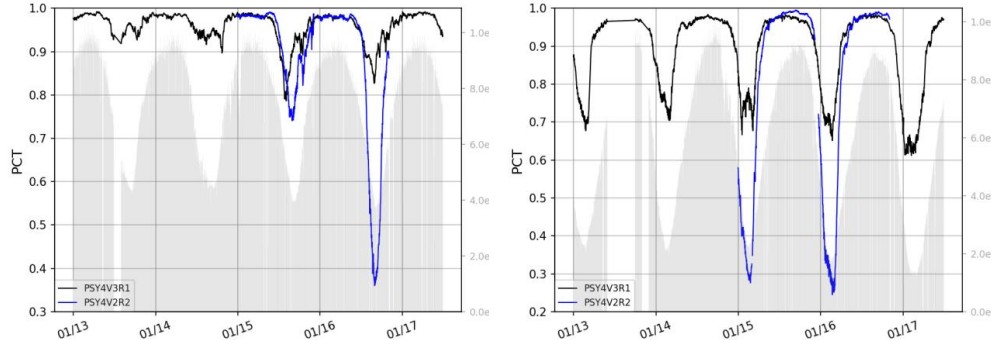

1    **Figure 28:** Time series of the PCT quantity for PSY4V2 (in blue) and PSY4V3 (in black). The left panel
2    corresponds to Arctic and the right panel to Antarctic. Time series of the number of available observations
3    appear in grey.





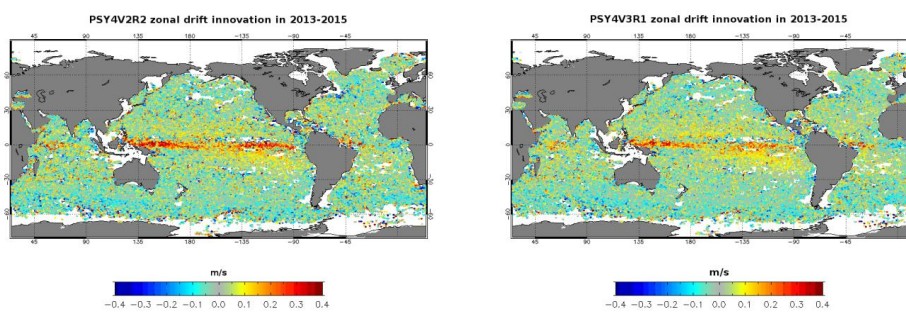

1   **Figure 29:** Mean zonal drift innovation (m s$^{-1}$) with PSY4V2 (on the left) and PSY4V3 (on the right) over the
2   time period 2013-2015. Observations come from Argo surface floats and a surface drifters corrected dataset
3   (Rio, 2012). Units are m s$^{-1}$.


| System acronym | Domain | Resolution | Model | Assimilation | Assimilated observations | MyOcean version | Mercator Ocean system reference |
|---|---|---|---|---|---|---|---|
| IRG_V0 | global | Horizontal: 1/4° Vertical: 50 levels | ORCA025 NEMO 1.09 LIM2, Bulk CLIO 24 h atmospheric forcing | SAM (SEEK) | "RTG" SST SLA T/S vertical profiles | V0 | PSY3V2R1 |
| IRG_V1V2 | global | Horizontal: 1/4° Vertical: 50 levels | ORCA025 **NEMO 3.1** **LIM2 EVP, Bulk CORE** **3 h atmospheric forcing** | SAM (SEEK) **IAU** **3D-VAR bias correction** | "RTG" SST SLA T/S vertical profiles | V1/V2 | PSY3V3R1 |
| IRG_V3V4 | global | Horizontal: 1/4° Vertical: 50 levels | ORCA025 NEMO 3.1 LIM2 EVP, Bulk CORE 3 h atmospheric forcing **New parameterization of vertical mixing** **Taking into account ocean colour for depth of light extinction** **Large scale correction to the downward radiative and precipitation fluxes** **Adding runoff for iceberg melting** **Adding seasonal cycle for surface mass budget** | SAM (SEEK) IAU 3D-VAR bias correction **Obs. errors higher near the coast (for SST and SLA) and on shelves (for SLA)** **MDT error adjusted** **Increase of Envisat altimeter error** **QC on T/S profiles** **New correlation radii** | **"AVHRR+AMSRE" SST** SLA T/S vertical profiles MDT "CNES-CLS09" adjusted **Sea Mammals T/S vertical profiles** | | PSY3V3R3 |

**Table 1:** Specifics of the Mercator Ocean IRG systems. In bold, the major upgrades with respect to the previous version. Available and operational production periods are described in Fig. 1.



| System acronym | Domain | Resolution | Model | Assimilation | Assimilated observations | MyOcean CMEMS version | Mercator Ocean system reference |
|---|---|---|---|---|---|---|---|
| HRG_V1V2 | global | Horizontal: 1/12° Vertical: 50 levels | ORCA12 NEMO 1.09 LIM2, Bulk CLIO 24 h atmospheric forcing | SAM (SEEK) IAU | "RTG" SST SLA T/S vertical profiles | V1/V2 | PSY4V1R3 |
| HRG_V3V4_V1V2 | global | Horizontal: 1/12° Vertical: 50 levels | ORCA12 NEMO 3.1 **LIM2 EVP, Bulk CORE** **3 h atmospheric forcing** **New parameterization of vertical mixing** **Taking into account ocean color for depth of light extinction** **Large scale correction to the downward radiative and precipitation fluxes** **Adding runoff for iceberg melting** **Adding seasonal cycle for surface mass budget** | SAM (SEEK) IAU **3D-VAR bias correction** **Obs. errors higher near the coast (for SST and SLA) and on shelves (for SLA)** **MDT error adjusted** **Increase of Envisat altimeter error** **QC on T/S profiles** **New correlation radii** | **"AVHRR+AMSRE" SST** SLA T/S vertical profiles MDT "CNES-CLS09" adjusted **Sea Mammals T/S vertical profiles** | V3/V4 V1/V2 | PSY4V2R2 |
| HRG_V2V3 | global | Horizontal: 1/12° Vertical: 50 levels | ORCA12 NEMO 3.1 LIM2 EVP, Bulk CORE 3 h atmospheric forcing New parameterization of vertical mixing Taking into account ocean colour for depth of light extinction Adding seasonal cycle for surface mass budget **50 % of model surface currents used for surface momentum fluxes** **Updated runoff from Dai et al, 2009 + runoff fluxes coming from Greenland and Antarctica** **Addition of a trend (2.2mm yr⁻¹) to the runoff** **Global steric effect added to the sea level** **New correction of precipitations using satellite data + no more correction of the downward radiative fluxes** **Correction of the concentration/dilution water flux term** **Relaxation toward WOA13v2 at Gibraltar and Bab-el-Mandeb** | SAM (SEEK) IAU **3D-VAR bias correction (1 month time window)** MDT error adjusted Increase of Envisat altimeter error QC on T/S profiles New correlation radii **Addition of a second QC on T/S vertical profiles** **Adaptive tuning of observation errors for SLA and SST** **New 3D observation errors files for assimilation of in situ profiles** **Use of the SSH increment instead of the sum of barotropic and dynamic height increments** **Global mean increment of the total SSH is set to zero** | **CMEMS OSTIA SST** SLA T/S vertical profiles **MDT adjusted based on CNES-CLS13** Sea Mammals T/S vertical profiles **CMEMS Sea Ice Concentration** **WOA13v2 climatology (temperature and salinity) constrain below 2000m (assimilation using a non-Gaussian error at depth)** | V2/V3 | PSY4V3R1 |

**Table 2:** Specifics of the Mercator Ocean HRG systems. In bold, the major upgrades with respect to the previous version. Available and operational production periods are described in Fig. 1.





|  | AMSR Ice | AMSR Water |
|---|---|---|
| Model Ice | Hit ice | False Alarm |
| Model Water | Miss | Hit water |

1   **Table 3:** Contingency table entries for sea ice verification of PSY4V3 system as compared to AMSR sea ice
2   concentration observations.

