# Peer review of "Recent updates on the Copernicus Marine Service global"

_Ocean Science, 2018_

## Referee Comment (RC1) · L. Vandenbulcke (Referee) · 12 Apr 2018

The paper entitled "Recent updates on the Copernicus Marine Service global ocean monitoring and forecasting real-time 1/12° high resolution system" presents the innovations in the near-real time global PSY4V3 system simulating 2006-present, compared to the previous system. The paper then validates the innovations by looking at essential ocean variables. The discussion of the improvements is generally convincing. The model itself is actually very convincing, and some of the examined metrics are truly impressive with very low errors. The paper is also very well written and clear.

[Figure]

I have a small list of remarks / questions, in no particular order:

- to facilitate the reading of the paper, please make the names of the systems consistent through the whole text and figures (PSY4V3 or HRG-V2V3, etc)

- section 3.2 about correction of precipitation. It seems that figure 6 represents ECMWF (left panel), and ECMWF corrected by PMWC (right panel). In this case, of course in the right panel, the differences compared to PMWC are smaller than in the left panel. Thus Fig 6 is not very relevant and could be removed. On the contrary, Fig 7 shows the impact of the correction, and is very convincing.

- section 3.3 about assimilating climatological profiles at depth, starts by giving a list of possible causes for this drift. Could you expand this a little? One would like to understand why below 2000m, the model would drift and present larger and larger biases over time (such as written page 16 around line 21, and illustrated in Fig 11a); as this is kind of surprising and one wonders if there is something that can be done to the model itself ?

- section 3.4, page 19 line 32, about the filtering of SLA anomalies, and the trapping of small structures, you say "this happens less" when filtering. Can you show this ? It is also surprising that there seems to be no clear advantage of 10 or 300 passes of the filter. Does this tell something about the spatial scales ?

- section 3.5, page 20, lines 5-8: you say the "error increases" when TIW are marked, and this can be explained by cloudy images or by the model shift of TIW. This could be more clear. I don't understand why images would be more clouded when TIW are more marked, is there any relation between these 2 processes (TIW and clouds). I agree with the second reason. In an ideal world, the error on model (resp. observations) would be determined without using the observations (resp. model); but the world is not ideal, and that the "Desrozieres" method is effective and hence should indeed be used to improve the model. Therefor, if the model shifts structures (such as TIW), one indeed may need to modify the error affecting observations. If this is what you meant, maybe somewhere in the section you could write such an introduction.

- for the whole of section 4, or maybe in section 1 or 2 already, you could specify clearly and once and for all if there are differences between the catch up period (2006-2016) and the operational period (2016-ongoing) ? I mean for atmospheric forcings, in situ observations reprocessing mode, etc. For example in section 4, at one point you say that in Jan 2014, you start using NRT observations (if I understood correctly) ? Does that mean that before 2014, you used reprocessed observations ?

- section 4.2 in particular seems to indicate that the system relies a lot on data. But this feeling is present in the whole paper. Actually almost all of the improvements to the PSY4V3 system seem to be data-based, whether they concern data assimilation, error modelisation, atmospheric forcing data (precipitation)... This is not a criticism, just something that I notice. Maybe it becomes extremely difficult to improve the model itself any further, apart from forcing it with more and better data.

- page 25 lines 1-5, you talk about 3.2 mm/year. But I understand from the previous sections that the mean SSH is not allowed to evolve freely, but is forced to increase 2 mm/year. Can you clarify this ?

Very general comments

- Among the errors that you noticed in section 4, is it possible to identify some culprit processes, where the model leads to biases that could potentially be corrected by better algorithms ? Maybe it's worth saying something about that in the paper.

- In the same line of ideas, it seems that the validation was done mostly on 2006-2016 (we don't see much about the NRT system, or maybe this is just a wrong impression I have). In particular, only for SLA do we see something about the lead time (1-7 days, so "3.5 days"). For other error metrics (SST RMS, etc), could it in the future be possible to assess the model as a function of lead time ? For example, when you give the SST rms ($0.1°C$ for example), and we speak about the NRT ("OPER") model, could you (a posteriori) compare observations with the forecast generated at day-1, day-2, ... , day-10, and give 10 values for the SST rms ?

Typos, language errors, and minor remarks:
- in general, the paper mixes direct and indirect styles "we do this ..." , "this was done..."
- page 2 line 24: "an"
- page 21 L28 : missing "The"
- page 22 L 27 : "as" -> "such as"
- page 23 L18 : 'worst" -> "worse"
- page 23 L23 : "after" -> "afterward"
- page 24 L18-20 : phrase is badly formulated
- page 25 : remove L20 (duplicates what's said just above)
- page 25 L 28: "solutionS"
- page 26 L5 : "from" -> "for"
- all figures are too small when printed on paper

---

## Referee Comment (RC2) · Anonymous Referee #2 · 22 Apr 2018

General comments This Discussion paper discusses the main updates of the Mercator Ocean operational forecasting systems at 1/12 resolution, which is the highest resolution deterministic forecast product released by CMEMS. The manuscript is certainly interesting and deserves publication because documents the main changes and quality increase achievements of a state-of-the-art oceanographic analysis system. It can be useful for both developers and users. However, in my opinion there are many scientific issues that are only empirically formulated, lack scientific justification and require a deeper explanation. In general, it is also not clear why some updates are discussed in details in section 3 and some others only mentioned in section 2: this seems quite arbitrary. I therefore recommend revising the manuscript to address the specific points

below to help the readership to understand the motivation and justification of such changes, which will really help the usefulness of this work in the oceanographic community and the robustness of the paper. I found the article well-written; it is a bit long, and I suggest the authors to consider removing some figures (29 figures are really too many in my opinion).

Specific comments

Abstract L23: forecast error → background error

Introduction P1L16: I believe the fact that Mercator is entrusted by EC is not relevant: here it is relevant that Mercator Ocean is in charge of the global analysis and forecast system

P1L27: "four many areas": I count 6 areas from the manuscript, moreover this number is subjective

P3L5-10: seems a repetition and suggest to merge in P2L14-26

P3L27: "three twin ..." the number three appear evident only later in the paper, suggest dropping it

P5L4-6: The point here is not that parameterizations in the version 3.1 of NEMO are still in the version 3.6, but how many new parameterizations and improvements of NEMO are you missing using version 3.1? In my opinion it should be discussed this way: although I perfectly understand that upgrading version is not easy for an operational system, and this is a justification for me, there are many years of ocean model developments not exploited here, which should be honestly mentioned.

P6L8: maybe is good to say what are the problems coming from the use of z-coordinate you are referring to? Would it be better then to use sigma-coordinates? Or you mean something else?

P6L22: it would be interesting to know what you found for 0, 50 and 100% of relative

wind and which was the criteria to choose 50%

P7L1: negative gridded anomalies : maybe better to say negative variations of water masses estimated from GRACE (if I interpret correctly)

P7L11: "...known..." suggest adding a reference

P7L32: I assume covariances are static (seasonal) but do not vary inter-annually in the real-time system. It is better to state it explicitly

P9 paragraphs starting at L15 and L19 seem in contradiction: if the obs errors are adaptive, why do you need a retuning?

P9L33: this requires a clarification on how you changed the formulation: from the anomaly dataset how do you define the SSH in the old and new system? Wind effect is also barotropic, i.e. is not clear what you actually changed

P10L12: suggest adding that the new approach is more consistent with what you actually do (using not a free run but a bias-corrected free run, which better mimics the operational system)

Section 2.3.1 & 2.3.2 It is not clear if these criteria are completely empirical or have some theoretical justification. If empirical as I guess, please discuss the criteria you used to obtain the values for the thresholds

P12L11 it seems weird that there are more suspicious obs in 2012 and 2013. Any idea why?

P13L1: how do you define the adjustment, achieved in 3 months?

Figure 5 and discussion. It looks like the new initialization bears more subsurface biases, so that it is not really convincing that it is better than the old one. I think it deserves a better discussion

Section 3.2: the discussion on the result (Figure 7) will benefit from a quantitative

assessment (RMSE and bias reduction of the model vs salinity obs at global scale will be sufficient)

Section 3.3 A reason for drift might be also inadequate background-error covariances that contain spurious correlations. This should be mentioned at the beginning. Again, the thresholds seem to be empirical and suggest writing the criteria for their adoption.

P18 Fig 12: It is weird that without the SEEK you have more variability than the observational product: I wonder whether the two datasets are really comparable, given that the 1/12 model may have a signal at higher resolution than the gridded altimeter product.

Section 3.4.2 Suggest putting it more in the context. I assume that the filtering is applied to the anomaly from the BIAS experiment before covariance computation, and then these differently filtered covariances are used in single-track experiment. However, it is a deduction and recommend to begin the section explaining this.

Section 3.5 This is the section that I found very hard to justify. I don't see a reasoning why observation errors are flow-dependent and should change so much with time, except the representativeness error component that might slightly change with season and/or particular events (eg presence of fronts, etc.). But this is less crucial than the background-errors that are certainly modulated by observation availability, large- and small scale processes, forcing, etc. This seems particularly true when looking at an observational dataset with nearly constant sampling (SST, Fig 18/19). I think the results improve not because observational errors really change with time, but because you are changing the ratio between background and observation errors, provided that background errors are of course flow-dependent as mentioned before. Moreover, the Desroziers method implies simultaneous tuning of background and observation errors. If the authors are able to provide a similar complementary retuning of background errors (I mean not with experiments but with diagnostics), it will really improve the robustness of the section. Otherwise a better discussion is needed, probably mentioning

that what is actually done is to change the relative weight between background and observation errors, rather than changing observation error themselves.

Figure 21: suggest better putting the figure and related discussion in context: the figure shows scores for assimilated vs non-assimilated (NOAA) datasets, so it is not clear the goal of the figure

Section 4.1.2 Title and text: as the SST source is similar between OSTIA and CAT-SAT (night time measurements from infrared sensors), I don't think the latter is really independent. I would define it "external" or similar Section 4.2 L21: 2005-2012 is not a decade; moreover, suggest trying the entire (inter-decadal) climatology to get rid of the weird increase of RMSE after 2012, probably due to the fact that the decadal mean is much too affected by the inter-annual variability therein.

Section 4.3.1 Please clarify how you estimate 2 and 4 cm for instrumental and MDT errors; the MDT one seems in particular arbitrary; also, in the computation of the statistics, do you use any threshold to filter out certain misfits, in order to obtain that global value of RMS, or you use all observations?

Section 4.3.2. Please provide a reference for BADOMAR and GLOSS/CLIVAR. Also in section 4.4.1 and 4.5 there are products that are referred to only through links: is there any better way to refer to them?

---

## Author Comment (AC1) · 1 Jun 2018

The paper entitled "Recent updates on the Copernicus Marine Service global ocean monitoring and forecasting real-time 1/12° high resolution system" presents the innovations in the near-real time global PSY4V3 system simulating 2006-present, compared to the previous system. The paper then validates the innovations by looking at essential ocean variables. The discussion of the improvements is generally convincing. The model itself is actually very convincing, and some of the examined metrics are truly impressive with very low errors. The paper is also very well written and clear.

We thank Luc Vandenbulcke (Referee #1) for his careful reading of our manuscript and for his constructive remarks. Following his advices, we tried to make the manuscript clearer. All remarks detailed below by the referee were considered and/or discussed.

I have a small list of remarks / questions, in no particular order:

- to facilitate the reading of the paper, please make the names of the systems consistent through the whole text and figures (PSY4V3 or HRG-V2V3, etc).

Figure 1, Table 1 and Table 2 have been modified and the text has been changed accordingly.

- section 3.2 about correction of precipitation. It seems that figure 6 represents ECMWF (left panel), and ECMWF corrected by PMWC (right panel). In this case, of course in the right panel, the differences compared to PMWC are smaller than in the left panel. Thus Fig 6 is not very relevant and could be removed. On the contrary, Fig 7 shows the impact of the correction, and is very convincing.

We think that Figure 6 allows to locate the areas of low and strong correction of precipitations and therefore the areas that potentially will be the most impacted. For this reason, we would like to keep this figure. The maps of the Figure 6 have also been re-centered on Pacific at it was already the case for Figure 7.

- section 3.3 about assimilating climatological profiles at depth, starts by giving a list of possible causes for this drift. Could you expand this a little? One would like to understand why below 2000m, the model would drift and present larger and larger biases over time (such as written page 16 around line 21, and illustrated in Fig 11a); as this is kind of surprising and one wonders if there is something that can be done to the model itself?

Referee #2 has also mentioned this point. The text has been completed and the following reference has been added:

De Lavergne C., Madec, G., Le Sommer, J., Nurser, A. G., and Naveira-Garabato, A. C.: On the consumption of Antarctic Bottom Water in the abyssal ocean, J. Phys. Oceanogr., 46, 635–651, doi:10.1175/JPO-D-14-0201.1, 2016.

- section 3.4, page 19 line 32, about the filtering of SLA anomalies, and the trapping of small structures, you say "this happens less" when filtering. Can you show this? It is also surprising that there seems to be no clear advantage of 10 or 300 passes of the filter. Does this tell something about the spatial scales?

We agree that the first version of the text was not very clear. Some sentences have been changed to clarify this point.

- section 3.5, page 20, lines 5-8: you say the "error increases" when TIW are marked, and this can be explained by cloudy images or by the model shift of TIW. This could be more clear. I don't understand why images would be more clouded when TIW are more marked, is there any relation between these 2 processes (TIW and clouds). I agree with the second reason. In an ideal world, the error on model (resp. observations) would be determined without using the observations (resp. model); but the world is not ideal, and that the "Desroziers" method is effective and hence should indeed be used to improve the model. Therefore, if the model shifts structures (such as TIW), one indeed may need to modify the error affecting observations. If this is what you meant, maybe somewhere in the section you could write such an introduction.

We agree with all of that. We added in section 2.2 that "only one SST map is assimilated on the fifth day of the 7-day cycle" and that "cloudy regions are filled by the analysis performed in OSTIA". We also added some supplementary explanations in section 3.5 making, we hope, the section clearer.

- for the whole of section 4, or maybe in section 1 or 2 already, you could specify clearly and once and for all if there are differences between the catch up period (2006-2016) and the operational period (2016-ongoing) ? I mean for atmospheric forcings, in situ observations reprocessing mode, etc. For example in section 4, at one point you say that in Jan 2014, you start using NRT observations (if I understood correctly)? Does that mean that before 2014, you used reprocessed observations?

The text has been completed in the introduction and in section 2.2 for all assimilated observations. For in situ temperature and salinity vertical profiles, it was already partially mentioned in the original manuscript page 9 (lines 7-14).

- section 4.2 in particular seems to indicate that the system relies a lot on data. But this feeling is present in the whole paper. Actually almost all of the improvements to the PSY4V3 system seem to be data-based, whether they concern data assimilation, error modelisation, atmospheric forcing data (precipitation)... This is not a criticism, just something that I notice. Maybe it becomes extremely difficult to improve the model itself any further, apart from forcing it with more and better data.

You are right. It is difficult to improve the model itself with the used version of NEMO.
We develop currently the next version of the global system, based on the version 3.6 of NEMO. Some new parameterizations present in this version will allow to improve the model itself (see first point in "Very general comments" part).

- page 25 lines 1-5, you talk about 3.2 mm/year. But I understand from the previous sections that the mean SSH is not allowed to evolve freely, but is forced to increase 2 mm/year. Can you clarify this?

The mean sea level time evolution is the result of an imposed trend for mass inputs (2.2 mm yr$^{-1}$, see section 2.1) together with a diagnostic steric effect re-computed from model temperature and salinity. Although the distribution between mass and steric diagnosed from the model is not yet fully satisfactory, the trend of the Global Mean Sea Level is consistent with the observations. This is already said in section 2.2 of the original manuscript.

The expected steric effect is about 1 mm yr$^{-1}$. The trend of the mean sea level corresponds to the sum of this steric effect and the trend of mass of 2.2 mm yr$^{-1}$.

Very general comments:

- Among the errors that you noticed in section 4, is it possible to identify some culprit processes, where the model leads to biases that could potentially be corrected by better algorithms? Maybe it's worth saying something about that in the paper.

You are right. We think also that some model biases could potentially be corrected by using better algorithms and more sophisticated parameterizations. Some of them are already available in NEMO 3.6 version. We added in the conclusion of the paper some sentences about the following algorithms/parameterizations we plan to use in the future system version:

- LIM3 multi-category ice model.
- Z* vertical coordinate, which basically consists in changing the total ocean thickness and the sea level accordingly. This leads to the relaxation of the linear free surface assumption approximation and allows for exact global tracers conservation and the removal of unphysical surface salt fluxes.
- Split explicit free surface in place of the actual filtered free surface of Roullet and Madec 2000. Apart from the better representation of external gravity waves, this also provides a substantial CPU gain on massively parallel architectures.
- Vertical mixing (transition from a one equation TKE closure to a two-equations GLS scheme, Reffray et al., 2014).
- A third order horizontal advection scheme (UBS Upstream Bias auto diffusive Scheme – Shchepetkin et al., 2005), replacing the second order vector form differencing for momentum.

Reffray, G., Bourdalle-Badie, R., and Calone, C.: Modelling turbulent vertical mixing sensitivity using a 1-D version of NEMO, Geosci. Model Dev. Discuss., 7, 5249-5293, http://www.geosci-model-dev.net/8/69/2015/gmd-8-69-2015.html, 2014.

Shchepetkin, A.F. and McWilliams, J.C.: The regional oceanic modeling system (ROMS): a split-explicit, free-surface, topography-following-coordinate oceanic model, Ocean Modelling, 9, 347-404, doi:10.1016/j.ocemod.2004.08.002, 2005.

- In the same line of ideas, it seems that the validation was done mostly on 2006-2016 (we don't see much about the NRT system, or maybe this is just a wrong impression I have). In particular, only for SLA do we see something about the lead time (1-7 days, so "3.5 days"). For other error metrics (SST RMS, etc), could it in the future be possible to assess the model as a function of lead time? For example, when you give the SST rms (0.1°C for example), and we speak about the NRT ("OPER") model, could you (a posteriori) compare observations with the forecast generated at day-1, day-2, ... , day-10, and give 10 values for the SST rms ?

You are right. We chose in this paper to show the impact of the many updates only on the hindcasts (catch up to real time which corresponds to the 2006-2016 period).

We performed also validation in NRT on forecasts and the performance of the daily 10-day forecasts has been checked. For instance, Figure A represents temperature RMS differences (model minus observation) for best analysis (hindcast) and for 1-day, 3-day, 5-day, 7-day and 9-day forecasts. As expected, the best analysis has the lower RMS and this RMS increases with the forecast length. Similar results are obtained for salinity, SLA and SST.

[Figure]

**Figure A:** Temperature (°C) RMS differences (model minus observation) in the 5-100m, 100-300m, 300-800m and 800-2000m layers. Statistics are displayed for best analysis (black line) and for 1-day (blue line), 3-day (red line), 5-day (green), 7-day (orange line) and 9-day (brown line) forecasts. The number of available observations appears in grey in the background.

Typos, language errors, and minor remarks:

- in general, the paper mixes direct and indirect styles "we do this ...", "this was done..."
- page 2 line 24: "an"
- page 21 L28 : missing "The"
- page 22 L 27 : "as" → "such as"
- page 23 L18 : "worst" → "worse"
- page 23 L23 : "after" → "afterward"
- page 24 L18-20 : phrase is badly formulated

- page 25 : remove L20 (duplicates what's said just above)
- page 25 L 28: "solutionS"
- page 26 L5 : "from" → "for"
- all figures are too small when printed on paper.

All these "typos" errors have been corrected. Some sentences have been reformulated. The size of the figures has also been increased.

---

## Author Comment (AC2) · 1 Jun 2018

General comments:

This Discussion paper discusses the main updates of the Mercator Ocean operational forecasting systems at 1/12 resolution, which is the highest resolution deterministic forecast product released by CMEMS. The manuscript is certainly interesting and deserves publication because documents the main changes and quality increase achievements of a state-of-the-art oceanographic analysis system. It can be useful for both developers and users.

We thank anonymous Referee #2 for his careful reading of our manuscript and for this comment.

However, in my opinion there are many scientific issues that are only empirically formulated, lack scientific justification and require a deeper explanation. In general, it is also not clear why some updates are discussed in details in section 3 and some others only mentioned in section 2: this seems quite arbitrary. I therefore recommend revising the manuscript to address the specific points below to help the readership to understand the motivation and justification of such changes, which will really help the usefulness of this work in the oceanographic community and the robustness of the paper.

The issues empirically formulated concern essentially the choice of the "threshold values". For those involved in the quality controls QC1 and QC2, the justification and the criteria for choosing the value of these parameters have been added in the text.

For those involved in section 3.3, we followed the tunings used by Greiner et al., 2008 (internal report) and we checked that these values allow the method to work properly. We give to the Referee the access to this report: https://cloud.mercator-ocean.fr/public.php?service=files&t=2f3c0f2d260b51aac32a4d03da71e2d3.

We also described in details only some of the updates mentioned in section 2. The choice of updates separately illustrated and discussed in section 3 may seem arbitrary. It corresponds in fact to the updates that doesn't result from routine system improvements (bathymetry, runoffs, assimilated databases, Mean Dynamic Topography, etc.). This is mentioned in the manuscript.

I found the article well-written; it is a bit long, and I suggest the authors to consider removing some figures (29 figures are really too many in my opinion).

We agree that. We think also that the number of figures is large. We tried to reduce it before submitting but, on the other hand, we believe that all figures are beneficial to understanding the system. This manuscript describes a complex system with a lot of new ingredients. A solution would be to split the paper in two parts: description of the system and details of the main updates (sections 2 and 3), and scientific assessment (section 4). This has not been our final choice, also wanting to measure, in the same paper, the overall impact of the integration of all updates on the products quality.

We would like therefore to keep all the figures.

Specific comments:
Following constructive comments, we tried to make the manuscript clearer and more detailed. All remarks detailed below by the referee were considered and/or discussed.

Abstract L23: forecast error → background error.
The text has been modified overall the manuscript.

Introduction P1L16: I believe the fact that Mercator is entrusted by EC is not relevant: here it is relevant that Mercator Ocean is in charge of the global analysis and forecast system.
We agree that. Only the relevant part has been kept in the text.

P2L27: "four main areas": I count 6 areas from the manuscript, moreover this number is subjective.
We added numbering of these four main areas (from (i) to (iv)) to better highlight them. This classification, and consequently the number of these main areas, is the one that appears in http://marine.copernicus.eu/markets/use-cases.

P3L5-10: seems a repetition and suggest to merge in P2L14-26.
We agree that. The text has been merged.

P3L27: "three twin ..." the number three appear evident only later in the paper, suggest dropping it.
We would like to maintain this paragraph in the introduction because we think that it is important to precise in the introduction that these three versions of system have been used to quantify the impact of the updates. We added some details about these simulations to clarify the paragraph.

P5L4-6: The point here is not that parameterizations in the version 3.1 of NEMO are still in the version 3.6, but how many new parameterizations and improvements of NEMO are you missing using version 3.1? In my opinion it should be discussed this way: although I perfectly understand that upgrading version is not easy for an operational system, and this is a justification for me, there are many years of ocean model developments not exploited here, which should be honestly mentioned.
We agree that. The text has been modified at the beginning of section 2.1: "The system PSY4V3 uses version 3.1 of the NEMO ocean model (Madec et al., 2008). This NEMO version is available since a few years and has been already used in the previous system PSY4V2. This was the available stable version of the code when we started the development of the system PSY4V3 a few years ago. Note that, using this version of the code, we do not access better algorithms and more sophisticated parameterizations present in the current NEMO 3.6 stable version that is now the standard version of the code."

P6L8: maybe is good to say what are the problems coming from the use of z-coordinate you are referring to? Would it be better then to use sigma-coordinates? Or you mean something else?
The following sentence has been added: "z-coordinates, compared to sigma, isopycnal or hybrid coordinates, induce excessive numerical mixing over overflow sills (Winton et al., 1998). Mediterranean overflow, without any relaxation, would settle at an equilibrium depth of 800 m or so otherwise instead of 1100 m observed. Sigma coordinates could indeed improve the

representation of overflow processes but are likely to induce other problems elsewhere due to sigma gradient pressure error over steep topography or excessive diapycnal mixing in the interior (Marchesiello et al., 2009)".

The two following references have been added:
Winton, M., R. Hallberg and A. Gnanadesikan, 1998: Simulation of Density-Driven Frictional Downslope Flow in Z-Coordinate Ocean Models. J. Phys. Oceanogr., 28, 2163-2174.
Marchesiello, P., L. Debreu and X. Coulevard, 2009: Spurious diapycnal mixing in terrain-following coordinate models: The problem and a solution. Ocean Modelling, 26 (3-4), 156-169.

P6L22: it would be interesting to know what you found for 0, 50 and 100% of relative wind and which was the criteria to choose 50%.
We followed the results obtained by "Bidlot J.R., 2012: Use of MERCATOR surface currents in the ECMWF forecasting system: a follow-up study, Research Department Memorandum R60.9/JB/1228, Internal report available on request".
In the conclusion of this report, it is written: "An impact study was performed with the ECMWF forecasting system in which surface currents from MERCATOR OCEAN were incorporated into the analysis as well as the forecast system. The data from MERCATOR were processed in such a way that only the slow varying features were retained. By prescribing surface current as part of the ocean surface boundary condition, it was demonstrated that both the surface stress and the surface wind profile above will adjust such that the effect on surface stress is **only about half** of what would have been intuitively obtained by subtracting the ocean current from the surface wind in which no account was taken of surface current."
We followed also what it was said in the slide 13 (Figure A) of this presentation: http://cersat.ifremer.fr/templates/cersat/resources/meetingTalks/1505_Bidlot.pdf

[Figure]

**Figure A:** Slide 13 of the presentation at Brest of Bidlot in 2012.

P7L1: negative gridded anomalies: maybe better to say negative variations of water masses estimated from GRACE (if I interpret correctly).

We agree that. The text has been modified following your suggestion. Moreover we have clarified the paragraph concerning the building of mean seasonal freshwater fluxes representing Greenland and Antarctica ice sheets and glaciers runoff melting.

P7L11: "...known..." suggest adding a reference.

References have been included.

P7L32: I assume covariances are static (seasonal) but do not vary inter-annually in the real-time system. It is better to state it explicitly.

The background error covariances in SAM rely on a fixed basis. They do not evolve in real-time but they contain the seasonal signal and the inter-annual signal from the 9-year simulation.

The following sentence has been added: "The background error covariances in SAM rely on a fixed basis, seasonally-variable ensemble of anomalies. They also contain the inter-annual signal from the 9-year simulation. This choice implies that, at each analysis step, a sub-set of anomalies is used to improve the dynamic dependency. A significant number of anomalies are kept from one analysis to the other (250 anomalies), thus ensuring error covariance continuity."

P9 paragraphs starting at L15 and L19 seem in contradiction: if the obs errors are adaptive, why do you need a retuning?

Adaptive tuning of errors has been implemented for satellite SLA and SST observations. The method has not been used for temperature and salinity vertical profiles because of the lack of in situ data. Three-dimensional fixed observation errors are then used for the assimilation of in situ temperature and salinity vertical profiles. It is mentioned in the paragraphs starting at L15 and L19 of the original manuscript and at the beginning of section 3.5.

P9L33: this requires a clarification on how you changed the formulation: from the anomaly dataset how do you define the SSH in the old and new system? Wind effect is also barotropic, i.e. is not clear what you actually changed.

You are right. We added some explanations in the text.

In the previous system PSY4V2, the SSH was split in barotropic and baroclinic components, as explained in Benkiran and Greiner, 2008 (page 2060). Moreover, in the system PSY4V2, barotropic height was computed without the wind effect.

The following reference has been added:
Benkiran, M. and Greiner, E.: Impact of the Incremental Analysis Updates on a Real-Time System of the North Atlantic Ocean, J. Atmos. Ocean. Tech., 25, 2055-2073, 2008.

P10L12: suggest adding that the new approach is more consistent with what you actually do (using not a free run but a bias-corrected free run, which better mimics the operational system).

We added a sentence as suggested by the reviewer.

Section 2.3.1 & 2.3.2 It is not clear if these criteria are completely empirical or have some theoretical justification. If empirical as I guess, please discuss the criteria you used to obtain the values for the thresholds.

We agree that. At the beginning of section 2.3, we added some explanations about the criteria we used to obtain the values of the thresholds.

P12L11 it seems weird that there are more suspicious obs in 2012 and 2013. Any idea why?
We agree that. The CORA 4.1 CMEMS in situ database includes the years 2012 and 2013 and we expected a percentage of suspicious profiles relatively stable until 2013. It is almost the case for the temperature profiles but not for salinity. It can not be connected to a strong ENSO event that could explain that more suspicious salinity profiles than usual are detected for instance in the tropical Pacific. We tried to see if it was related to a network effect, but it is not the case. We asked also to the database producers and they have no explanation.

P13L1: how do you define the adjustment, achieved in 3 months?
We consider that an acceptable adjustment is achieved when between 80% and 90% of global energy is reached. To illustrate that, Figure B shows the evolution of the percentage of the three monthly quantities: turbulent kinetic energy (TKE), mean kinetic energy (MKE) and eddy kinetic energy (EKE). For all quantities, 90% of global energy is reached after 6 months. So we changed in the text "3 months" by "6 months". It does not change the discussion.

[Figure]

**Figure B:** Evolution of the percentage of the three monthly quantities: turbulent kinetic energy (TKE), mean kinetic energy (MKE) and eddy kinetic energy EKE. The 100% percentage corresponds to the mean of the months 9 to 20.

Figure 5 and discussion. It looks like the new initialization bears more subsurface biases, so that it is not really convincing that it is better than the old one. I think it deserves a better discussion. The text has been modified.

Section 3.2: the discussion on the result (Figure 7) will benefit from a quantitative assessment (RMSE and bias reduction of the model vs salinity obs at global scale will be sufficient).

We agree that. The text has been completed.

Section 3.3: A reason for drift might be also inadequate background-error covariances that contain spurious correlations. This should be mentioned at the beginning. Again, the thresholds seem to be empirical and suggest writing the criteria for their adoption.

Referee #1 has also mentioned this point. The text has been modified.

Regarding the value of the thresholds, we followed the tunings used by Greiner et al., 2008 and we checked that these values allow the method to work properly for PSY4V3.

P18 Fig 12: It is weird that without the SEEK you have more variability than the observational product: I wonder whether the two datasets are really comparable, given that the 1/12 model may have a signal at higher resolution than the gridded altimeter product.

We try to make the two datasets comparable by subsampling the 1/12° model (1 point every 3) before doing the comparison with DUACS which is a product on a ¼° regular horizontal grid. This has been clarified in the text.

Also here is an explanation regarding the excess of energy present in the BIAS simulation compared to the observational DUACS product. Figure C shows the mean currents on October-November-December 2013 with superimposed stream lines for the three simulations (FREE, BIAS, OPER) in the zoom (175° W – 125° W / 65° S – 20° S).

[Figure]

**Figure C:** Mean currents on October-November-December 2013 with superimposed stream lines for the three simulations (FREE, BIAS, OPER) in the zoom (175° W – 125° W / 65° S – 20° S).

South of 50° S, the Polar Front (PF) is more pronounced in BIAS and even more in OPER. On the other hand, between the East Australian Current (EAC) and the PF, the meridian gradient decreases in OPER. The gradient is not well maintained in BIAS north of the PF. The front leaks to the north in the less energetic zone where we thus find "spurious" meanders and eddies.

The average of the currents shows that the FREE has two very marked veins on the southern edges of the EAC and on the northern edge of the PF. This prevents the export of vorticity (which is advected zonally). In BIAS, the edges are less marked, and there are veins of current towards the less energetic zone. These veins disappear in OPER, especially south of the EAC. It can be noted that the BIAS shows PF connection at 150° W/48° S and EAC connection at 148° W / 35° S (black arrows).

To reduce that in the future, we plan to increase viscosity model coefficient or temperature and salinity in situ observations errors (or both) for the simulation using only temperature and salinity 3D-VAR large-scale biases correction.

Section 3.4.2: Suggest putting it more in the context. I assume that the filtering is applied to the anomaly from the BIAS experiment before covariance computation, and then these differently filtered covariances are used in single-track experiment. However, it is a deduction and recommend to begin the section explaining this.

We have switched the two subsections of section 3.4 to make it clearer. We have also better introduced these two subsections in the introduction of the section 3.4.

Section 3.5: This is the section that I found very hard to justify. I don't see a reasoning why observation errors are flow-dependent and should change so much with time, except the representativeness error component that might slightly change with season and/or particular events (eg presence of fronts, etc.). But this is less crucial than the background-errors that are certainly modulated by observation availability, large- and small scale processes, forcing, etc. This seems particularly true when looking at an observational dataset with nearly constant sampling (SST, Fig 18/19). I think the results improve not because observational errors really change with time, but because you are changing the ratio between background and observation errors, provided that background errors are of course flow-dependent as mentioned before. Moreover, the Desroziers method implies simultaneous tuning of background and observation errors. If the authors are able to provide a similar complementary retuning of background errors (I mean not with experiments but with diagnostics), it will really improve the robustness of the section. Otherwise a better discussion is needed, probably mentioning that what is actually done is to change the relative weight between background and observation errors, rather than changing observation error themselves.

We agree that. When we say "tuning of observations errors", we mean the sum of the instrumental and representativeness errors. It's true that the instrumental error doesn't change with time. On the contrary, the representativeness error is really flow-dependent.

We tried to apply the "Desroziers method" simultaneous on background and observation errors. But both errors tend to increase or decrease together. This evolution is slow but it is regular and

meaningless regarding the true errors. The ratio between background and observation errors remains constant. Moreover, in the OPER simulation and as mentioned in Lellouche et al. (2013) in the description of the data assimilation system SAM, an adaptive scheme corrects the background variance and gives an optimal background error variance based on a statistical test formulated by Talagrand (1998). This is why we let "Desroziers method" to adjust "instrumental + representativeness" error and "Talagrand method" to adjust the background error.
The text has been modified to make it clearer.

Figure 21: suggest better putting the figure and related discussion in context: the figure shows scores for assimilated vs non-assimilated (NOAA) datasets, so it is not clear the goal of the figure.
The text in section 4.1.1 has been completed.

Section 4.1.2 Title and text: as the SST source is similar between OSTIA and CATSAT (night time measurements from infrared sensors), I don't think the latter is really independent. I would define it "external" or similar.
We changed it.

Section 4.2 L21: 2005-2012 is not a decade; moreover, suggest trying the entire (inter-decadal) climatology to get rid of the weird increase of RMSE after 2012, probably due to the fact that the decadal mean is much too affected by the inter-annual variability therein.
"2005-2012 decade" has been changed to "2005-2012 truncated decade". The five previous decades of WOA13v2 monthly climatology from 1955 and that can be found on the NODC website, properly represent 10-year periods.
It is true that the "2005-2012 truncated decade" contains strong La Niña event (2010-2011) and, as a consequence, is biased to cold. The previous decades (before 2005) are even colder and can no longer be used for recent dates. Moreover, 2005-2012 "truncated decade" doesn't contain the period of transition towards El Niño events and in particular the strong one occurring in 2015. This explains the increase of RMS difference between the WOA13v2 monthly climatology and the in situ observations after 2012. Using the entire (inter-decadal) climatology will globally increase this RMS from 2007 to 2012 but the "weird increase" after 2012 will be still present even if it will be a little reduced.
We clarified the text of section 4.2.

Section 4.3.1 Please clarify how you estimate 2 and 4 cm for instrumental and MDT errors; the MDT one seems in particular arbitrary; also, in the computation of the statistics, do you use any threshold to filter out certain misfits, in order to obtain that global value of RMS, or you use all observations?
The 2 cm is the instrumental prescribed error. It is the error value recommended by data centers.
We prescribe also in the system an a priori MDT error (Figure D), which is equal to 4 cm in average on the regions observed by altimetry.
The text has been modified.

[Figure]

**Figure D:** MDT error a priori prescribed in the system PSY4V3.

Section 4.3.2. Please provide a reference for BADOMAR and GLOSS/CLIVAR. Also in section 4.4.1 and 4.5 there are products that are referred to only through links: is there any better way to refer to them?

References have been added for BADOMAR product and GLOSS tide gauge stations in section 4.3.2.

The link in section 4.5 has been replaced by a more classical reference. It concerns the Quality Information Document (QuID) for the product in question, which can be accessed via the CMEMS catalogue.

The link in section 4.4.1 has not been replaced because no evident classical reference has been found.

---

## Referee Report (RR1)

This is my second review of the manuscript OS-2018-15. I acknowledge that the authors have responded and incorporated all my previous comments in the revised version,
and I therefore think the manuscript can be published after a very few corrections are considered by the authors.

P2 Line 25: "and is in charge of the global high resolution ocean analyses and forecasts" is a repetition of Line 19-20

P4L3 Suggest rephrasing:
"was run ... to catch-up the real-time analysis and forecast system by ingesting "reprocessed" input data (atmospheric forcing, observations, etc.)"
I think the word "databases" is too vague in this context

L23
"current NEMO 3.6 stable version that is now the standard version of the code. "

I think it is better to say "the latest official release of NEMO" instead of "stable", "current" or "standard"

P9L7: It is clearer now, however it would be better to state explicitly that the inter-annual signal of the background errors is present only in the hindcast (reprocessed system) and not in the real time system, if I understand correctly. Suggest indeed writing clearly the differences between these two systems

Is the QC1 (Equation 1 ) able to retain observations with large innovations but far from climatology? (in case of regimes far from climatology) If so, suggest stating it clearly

P15L21 ", whose motions"

Section 3.4.1
I think the title itself is not clear, what is anomaly in this context? Innovations or analysis increments? It is important to make clear if the filtering is applied to the assimilation inputs (innovations) or outputs (analysis increments)

Section 3.4.1
If there is a reference about effective resolution of DUACS, would be good to insert it

Change the occurrences of "doesn't/don't" to "does not/do not"

---

## Author Response (AR2)

This is my second review of the manuscript OS-2018-15. I acknowledge that the authors have responded and incorporated all my previous comments in the revised version, and I therefore think the manuscript can be published after a very few corrections are considered by the authors.

All remarks detailed below by the referee were considered.

In the new revised manuscript, the previous corrections (previous revised manuscript) remain in red and the new ones are in blue.

P2 Line 25: "and is in charge of the global high resolution ocean analyses and forecasts" is a repetition of Line 19-20.

The sentence has been changed.

P4L3 Suggest rephrasing: "was run ... to catch-up the real-time analysis and forecast system by ingesting "reprocessed" input data (atmospheric forcing, observations, etc.)"
I think the word "databases" is too vague in this context.

The sentence has been rephrased.

P5L23
"current NEMO 3.6 stable version that is now the standard version of the code."
I think it is better to say "the latest official release of NEMO" instead of "stable", "current" or "standard".

We changed it.

P9L7: It is clearer now, however it would be better to state explicitly that the inter-annual signal of the background errors is present only in the hindcast (reprocessed system) and not in the real time system, if I understand correctly. Suggest indeed writing clearly the differences between these two systems.

We clarified the time period over which the anomalies were calculated and we added this sentence in the text: "Currently, the anomalies used in real time come from the set of anomalies computed over the 2007-2015 period with no real time extension of this set. We therefore make the hypothesis that the set of anomalies computed over a period prior to real time is able to represent correctly the background error covariance over the real time period."

Is the QC1 (Equation 1) able to retain observations with large innovations but far from climatology? (in case of regimes far from climatology). If so, suggest stating it clearly.

Yes, the QC1 is able to retain observations with large innovations but far from climatology. It was already stated in the previous corrected manuscript (see section 2.3.1, page 12, lines 22-26).

P15L21: "whose motions".

We changed it.

Section 3.4.1
I think the title itself is not clear, what is anomaly in this context? Innovations or analysis increments? It is important to make clear if the filtering is applied to the assimilation inputs (innovations) or outputs (analysis increments).

We added in the first paragraph of section 3.4.1 the following sentence: "Another way to remove the very short scales would be to filter the analysis increments before injecting them into the model. This choice would have led to a less optimal analysis and to a loss of balance between the different components of the increment."

Moreover, we have clarified at the end of the section 2.2 that the anomalies were inputs of the analysis.

Section 3.4.1
If there is a reference about effective resolution of DUACS, would be good to insert it.
This was already inserted in the previous corrected manuscript (see section 3.4.2, page 21 - line 33).

Change the occurrences of "doesn't/don't" to "does not/do not"
We changed it.

[revised manuscript text omitted]